# The contribution of X-linked coding variation to severe developmental disorders

Hilary C. Martin [1✉], Eugene J. Gardner [1], Kaitlin E. Samocha [1], Joanna Kaplanis[1], Nadia Akawi[1,2], Alejandro Sifrim[1,3], Ruth Y. Eberhardt [1], Ana Lisa Taylor Tavares [1,4,5], Matthew D. C. Neville [1], Mari E. K. Niemi [1,6], Giuseppe Gallone[1,7], Jeremy McRae[1,8], Deciphering Developmental Disorders Study*, Caroline F. Wright [9], David R. FitzPatrick[10], Helen V. Firth[1,4] & Matthew E. Hurles [1]

Over 130 X-linked genes have been robustly associated with developmental disorders, and X-linked causes have been hypothesised to underlie the higher developmental disorder rates in males. Here, we evaluate the burden of X-linked coding variation in 11,044 developmental disorder patients, and find a similar rate of X-linked causes in males and females (6.0% and 6.9%, respectively), indicating that such variants do not account for the 1.4-fold male bias. We develop an improved strategy to detect X-linked developmental disorders and identify 23 significant genes, all of which were previously known, consistent with our inference that the vast majority of the X-linked burden is in known developmental disorder-associated genes. Importantly, we estimate that, in male probands, only 13% of inherited rare missense variants in known developmental disorder-associated genes are likely to be pathogenic. Our results demonstrate that statistical analysis of large datasets can refine our understanding of modes of inheritance for individual X-linked disorders.

[1] Wellcome Sanger Institute, Wellcome Genome Campus, Hinxton, UK. [2] Division of Cardiovascular Medicine, Radcliffe Department of Medicine, University of Oxford, Oxford, UK. [3] Department of Human Genetics, University of Leuven, Leuven, Belgium. [4] Department of Clinical Genetics, Cambridge University Hospitals NHS Foundation Trust, Cambridge, UK. [5] Genomics England, Queen Mary University of London, London EC1M 6BQ, UK. [6] Institute for Molecular Medicine Finland, University of Helsinki, Tukholmankatu 8, Helsinki FI-00014, Finland. [7] Max Planck Institute for Molecular Genetics, Ihnestraße 63, 14195 Berlin, Germany. [8] Illumina Inc., 5200 Illumina Way, San Diego, CA 92122, USA. [9] Institute of Biomedical & Clinical Science, University of Exeter Medical School, Exeter EX2 5DW, UK. [10] MRC Human Genetics Unit, MRC IGMM, University of Edinburgh, Western General Hospital, Edinburgh EH4 2XU, UK. *A list of authors and their affiliations appears at the end of the paper. ✉email: hcm@sanger.ac.uk

Several attributes of X-chromosomal biology render it unique among chromosomes, and have profoundly influenced the landscape of X-linked monogenic disorders. The hemizygosity of the X chromosome in males results in a distinctive male-specific pattern of segregation in pedigrees for X-linked recessive disorders, which has facilitated the recognition of such disorders and catalysed the identification of the underlying associated genes[1]. By contrast, X-linked dominant disorders do not result in such characteristic segregation patterns in pedigrees, and are expected predominantly in females due to the considerably lower mutation rate of the maternally-inherited X chromosome in males[2].

While most X-linked disorders exhibit a profound sex-bias, suggestive of the underlying mode of inheritance, it is frequently observed that both sexes can manifest the same disorder. There are several possible explanations, which are not mutually exclusive. Skewing of X chromosome inactivation in females (which normally achieves dosage compensation) provides a mechanism for some female carriers of pathogenic variations in X-linked recessive genes to manifest disease of varying severity levels, although extreme skewing is rare on a population level[3,4]; this mechanism has previously been inferred to occur in 7.6% of female patients with intellectual disability[5]. One alternative explanation is an incompletely penetrant dominant phenotype associated with a variant that is fully penetrant when hemizygous; a special case of this is semi-dominance, in which heterozygous females are affected but hemizygous males are seldom observed due to lethality. Another explanation is that a disorder is truly dominant such that the hemizygous and heterozygous phenotypes are identical. These complexities have led some to recommend that the field refer collectively to 'X-linked disorders', avoiding explicit classification based on their individual modes of inheritance[6,7]. Nonetheless, X-linked Mendelian disease genes are often classified as 'X-linked recessive' (XLR) or 'X-linked dominant' (XLD) e.g. by the Online Mendelian Inheritance in Man catalog (OMIM; https://omim.org/).

The majority of X-linked monogenic disorders that have been identified are DDs, especially neurodevelopmental disorders such as intellectual disability (ID). The highly-curated DDG2P (Developmental Disorder Gene-to-Phenotype) database[8] contains over 130 DD-associated genes on the X chromosome, most of which are observed predominantly in males and are presumed X-linked recessive disorders, which can be caused by either de novo mutations or maternally inherited variants. Analyses of new large datasets of population variation have called into question a few of these gene associations[9], although most remain robust. Importantly, mutations in the same gene can cause more than one condition: 39 of the 132 X-linked genes in DDG2P are associated with more than one syndrome, many of them with different modes of inheritance and different mechanisms (e.g. loss-of-function versus activating).

The recent availability of exome sequencing, large cohorts of both cases and controls, and a fine-grained understanding of the germline mutation rate[10], have together empowered 'burden' analyses which can quantify the absolute and relative contributions of different classes of inherited and de novo variation to particular disorders and subsets of patients[11–14]. It has been suggested that the 1.3-fold male bias in the incidence of ID can be largely attributed to the male-biased contribution of X-linked disorders[15,16], although this has not been formally demonstrated. Here we analyse exome sequencing data from 11,044 families in the Deciphering Developmental Disorders (DDD) study and show that the relative contribution of X-linked causes of DDs is similar in male and female probands. We estimate the relative contribution of de novo versus inherited pathogenic variants in males (finding 41% de novo overall, 36% in X-linked recessive genes) and explore positive predictive values of different classes of variants, showing that this is very low for inherited missense variants in males. Furthermore, we develop an improved method to detect X-linked disease genes which identifies 23 genes, all of which have already been associated with DDs. Our study demonstrates the value of large cohort-based burden analyses for informing clinical practice and refining our understanding of inheritance modes for X-linked disorders.

## Results

**Comparison of male versus female phenotypes in DDD.** There are 40% more male than female probands in the DDD study (7844 males versus 5618 females), similar to the bias reported in other ID/DD cohorts[9], but lower than the four-fold male bias reported in autism cohorts[17]. We compared the phenotypes of male versus female probands in the study to explore whether phenotypic differences might be contributing to recruitment bias in males. Males were more likely to have another affected family member than females (26.5% versus 21.0%; Fisher's exact test $p = 5 \times 10^{-13}$). They tended to be recruited ~4.8 months earlier than females (linear regression $p = 0.0004$), so we controlled for age at assessment in the following tests of phenotypic differences. Males had slightly more affected organ systems than females, although this was only nominally significant (mean and ranges: 3.55 [1–12] for males, 3.49 [1–11] for females; linear regression $p = 0.049$; Supplementary Fig. 1). There were significant differences in the prevalence of several phenotypic features between the sexes (Supplementary Data 1). For example, after correcting for age at assessment, males were 2.4-times more likely to have an abnormality of the genitourinary system (logistic regression $p = 1.3 \times 10^{-48}$), 2.1-times more likely to have autistic behaviour ($p = 8 \times 10^{-41}$), and 2.0-times more likely to show hyperactivity ($p = 7 \times 10^{-10}$). However, none of these differences were large enough in magnitude to suggest that they made a major contribution to recruitment bias. Males in DDD were significantly taller (linear regression $p = 8 \times 10^{-8}$) and had greater occipital frontal circumference at recruitment ($p = 4 \times 10^{-17}$) than females (measured relative to the sex- and age-adjusted distributions in the general population) (Supplementary Table 1). Curiously, we observed that males walked on average 1.2 months earlier than females in the study (linear regression $p = 2 \times 10^{-7}$) (Supplementary Table 1), although in the general population, they tend to walk about two weeks later than females on average[18]. This may correspond with the observation that females in the cohort are slightly more likely than males to have severe ID/DD (12.4% versus 10.9%; logistic regression $p = 0.02$) but equally likely to have mild or moderate ID/DD (Supplementary Data 1). These comparisons suggest that although there are some significant differences in average phenotypes between the sexes in DDD, these are small in magnitude and males and females are broadly similar in clinical presentation.

**X chromosome burden analysis.** We hypothesised that the higher number of males in DDD might be due to variation on the X chromosome, since males, being haploid, might be more vulnerable to pathogenic variants on this chromosome and could be affected by both de novo and maternally inherited variants. To avoid biases that would be introduced by considering only diagnoses in known DD genes, we carried out a sex-specific burden analysis to estimate the fraction of patients attributable to rare or de novo coding variants in all genes in the non-pseudoautosomal regions of the X chromosome, assuming a monogenic model with full penetrance. We focused on 7136 independent male probands (5138 in family trios) and 3908 independent female probands in trios, and considered variants

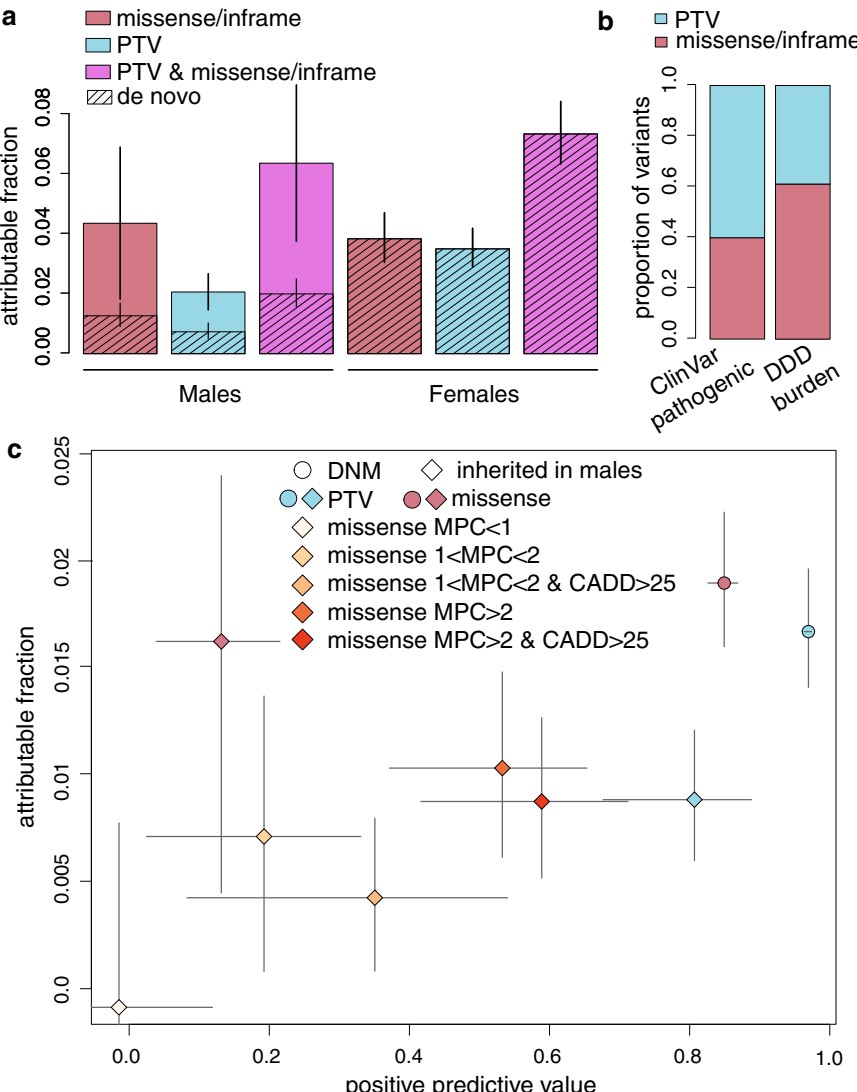

**Fig. 1 Results from burden analysis of rare and de novo coding variants on the X chromosome. a** Fraction of males and females attributable to rare inherited and de novo coding variants on the X chromosome. Note that in the males, the overall attributable fraction was estimated from the case/control analysis of all male probands (7136 cases versus 8551 controls), whereas that for de novo mutations (DNMs) was estimated only in the 5138 male trio probands. In the females, only de novos were considered since we were assuming full penetrance. **b** Relative fraction of protein-truncating variants (PTVs) versus missense/inframe variants amongst ClinVar likely pathogenic or pathogenic variants in X-linked DDG2P genes, versus the fraction inferred in the burden analysis in DDD. **c** Estimated attributable fraction versus positive predictive value for DNMs and inherited variants in males in X-linked DDG2P genes. Inherited missense variants are split according to CADD (Combined Annotation Dependent Depletion) 21 and MPC (missense badness, PolyPhen, constraint) 22 scores. In **a**, **c** the coloured bars (**a**) or points (**c**) show the point estimate and the error bars show 95% confidence intervals calculated as described in the 'Methods'.

with minor allele frequency <0.1% and with no hemizygotes in the gnomAD resource of population variation[19]. In male probands, we performed a case/control analysis, comparing probands to 8551 unaffected DDD fathers. This analysis implicitly includes both inherited variants and DNMs. For the purposes of some analyses, we also conducted an enrichment analysis of DNMs alone in the male trio probands, with quality control optimised to detect DNMs (Supplementary Data 2), and compared to a null mutation model[10]. In female trio probands, we performed a DNM enrichment analysis, assuming that there would be very few inherited pathogenic variants on the X chromosome in females since the vast majority of parents are unaffected. We are thus assuming full penetrance in females.

Overall, we estimated from the burden analysis that 6.0% of males (95% confidence interval [3.6–8.6%]) and 6.9% of females

[5.9–7.9%] had a pathogenic X-linked protein-truncating or missense/inframe variant (Fig. 1a). This implies that monogenic X-linked coding causes of DDs are not the cause of the male bias in DDD. In females, 90% [83–97%] of the burden was in DD-associated genes, versus 63% [44–100%] in males (95% confidence intervals from bootstrapping shown in Supplementary Fig. 2A, B). In trio males, 41% [23–100%] of the burden was de novo and the rest inherited (Supplementary Fig. 2C). Of the 127 de novo PTVs or missense/inframe mutations observed in males, eight (6%) appeared mosaic in mothers, and 12 (9%) post-zygotic mosaic in the probands.

The relative contribution of missense/inframe variants was higher in males than females, but not significantly so (Fig. 1a). Overall, 38.9% of the X-linked exonic burden was driven by protein-truncating variants (PTVs), with the rest being missense/

inframe (Fig. 1b). In contrast, in a set of 3906 variants in X-linked DD-associated genes reported as 'pathogenic' or 'likely pathogenic' in ClinVar[20], 60.6% were PTVs and 39.4% were missense/inframe variants. This is significantly different from our burden analysis of DD-associated X-linked genes (Fisher's exact test $p = 5 \times 10^{-21}$; Fig. 1b), and likely reflects the fact that PTVs are easier to interpret and hence more likely to be considered pathogenic or likely pathogenic by clinical geneticists and genetic diagnostic laboratories.

We next estimated the fraction of the observed rare inherited or de novo variants in known X-linked DD genes were actually likely to be pathogenic (i.e. the positive predictive value, PPV) (Fig. 1c). These PPVs can assist accurate diagnostic interpretation by providing prior probabilities of pathogenicity for different classes of variation. For de novo PTVs, de novo missense mutations and inherited rare PTVs in males, the PPV was >80%. However, the PPV for inherited rare missense variants (MAF < 0.001) that were not observed as hemizygotes in gnomAD was estimated to be only 13.2% [3.9–21.5%], indicating that there is a substantial risk of incorrect diagnosis and hence clinical mismanagement. In line with this, of variants passing these filters that have been reported in DECIPHER and rated by clinicians, 56/272 (20.6% [15.9–25.9%]) were classed as pathogenic or likely pathogenic, and 47.8% as 'uncertain' [41.7–53.9%]. If we did not apply the additional filter requiring zero hemizygotes in gnomAD in our burden analysis, the PPV was not significantly different from 0, although it increased to 15.6% [5.4–24.6%] if we reduced the MAF filter from 0.1% to 0.005% (Supplementary Fig. 3). We were able to increase the PPV to ~60% by applying more stringent filters on CADD[21] and MPC[22] scores to the missense variants (Fig. 1c), indicating that such filters are likely to aid clinical genetics practice by reducing rates of incorrect assignment of pathogenicity. However, Fig. 1c also shows that this improvement in specificity is counterbalanced by some reduction in sensitivity. In line with this, of 56 inherited X-linked inherited variants that have been reported in DECIPHER and rated pathogenic or likely pathogenic by clinicians, 52 (93%) had MPC > 1 and 38 (68%) had MPC > 2, whereas the PPVs for variants with these filters were 31% and 53% respectively. These results indicate the value of this population-based burden analysis for informing improvements in clinical practice.

**Contribution of de novo versus inherited variants in X-linked recessive genes in males.** In 1935, Haldane showed that the relative contribution of de novo versus inherited variants in X-linked recessive genes is a function of the reproductive fitness of a disorder and the mutation rate in the paternal and maternal germline[23]. Specifically, the fraction of male X-linked recessive cases due to DNMs should be $\frac{m\mu}{2\mu+\nu}$, where $m$ is the reproductive loss in affected males, $\mu$ is the mutation rate in eggs and $\nu$ is the mutation rate in sperm. For DDs which are reproductively lethal (i.e. $m = 1$), if the maternal and paternal germline mutations rates were equal, one-third of pathogenic variants would be expected to be de novo. However, the paternal mutation rate of single nucleotide variants (SNVs), the predominant class of pathogenic variant, is ~3.5 times higher than the maternal rate[24], which would be expected to lower the proportion of pathogenic SNVs that arise de novo to ~18% (see 'Methods').

We tested Haldane's theory by evaluating the fraction of the estimated SNV burden (observed-expected) in X-linked recessive genes in males that was de novo. We found that this fraction was ~36%, although, even within this large dataset, the bootstrap confidence intervals around this estimate remain large ([19–83%]) (Supplementary Fig. 2D). Using an approximation to a binomial distribution, we estimate that the probability of

observing our data under the null hypothesis of Haldane's theory is about 0.0001 (two-sided $p$-value from a two-sample test for equality of proportions). Together with our results from bootstrapping, this suggests that our data are unlikely to be consistent with Haldane's theory. There are at least three non-mutually exclusive explanations for this. Firstly, the DDD study may be biased away from classic inherited X-linked families because these are easier to diagnose through the usual clinical means and because patients who had been previously recruited to the UK-wide GOLD study focused on X-linked ID[1] tended not to be recruited to DDD. Secondly, it may be that our assumption about the ratio of male to female mutation rates differs on the X chromosome compared to the autosomes; however, the male:female mutation rate ratio would have to be substantially lower on the X chromosome than the autosomes to be consistent with the observed 36% DNMs, which seems unlikely. Finally, Haldane's theory did not incorporate the possibility of a reduced number of offspring in heterozygous carrier females, which would be expected to increase the proportion of DNMs in XLR genes. We note that this need not be due to any physiological phenotype in carrier females, and could include the effect of women choosing not to have more children after having one or more affected sons. This phenomenon is well recognised in current clinical practice and can result in halving the number of offspring of women known to be at risk of being carriers[25].

We tested the effect of rare PTVs in XLR genes in females in UK BioBank ($N = 13$ carriers; Supplementary Table 2) and found that carrier females had a nominally significant reduced number of children (average uncorrected values: 1.31 for carriers, 1.76 for non-carriers; ratio $t$-test $p = 0.038$), with a fertility ratio of carriers to noncarriers of 0.742 [0.503–0.981] (Supplementary Fig. 4). Shermer et al.[26] built on Haldane's theory and determined that the expected fraction of male diagnoses in XLR genes that are de novo should be $\frac{\mu(2f+m-mf)}{2\mu+\nu-\nu f}$, where $f$ is the reproductive loss in female carriers, $m$ the reproductive loss in affected males (assumed to be 100% here), $\mu$ is the mutation rate in eggs and $\nu$ in the mutation rate in sperm (assumed to be equal to $3.5\,\mu$ here). Thus, the level of reduced fertility observed in female carriers in UK BioBank implies we should expect 27.3% of the burden in X-linked recessive genes to be de novo, and this fraction could be as high as 39.8% considering the lower bound of the fertility ratio. This fraction overlaps the 95% confidence interval we estimate from bootstrapping ([19–83%]; Supplementary Fig. 2D). Hence, reduced numbers of offspring in female carriers may be contributing to this fraction being higher than expected under Haldane's theory.

**Gene discovery and delineation of inheritance mechanisms.** Our previous efforts at gene discovery on the X chromosome involved testing for de novo enrichment in males and females combined[12]. This is best powered to detect XLD genes but ignores the substantial contribution from inherited variants in males, so will be underpowered to find new XLR genes. In contrast, many previous gene discovery studies focused on males from obviously X-linked pedigrees so will have missed de novo causes of X-linked disorders[1]. Hence, we implemented three different tests to optimise power to detect XLR, XLD, and X-linked semi-dominant (male lethal) genes (see Methods). This identified 23 genes that passed Bonferroni correction (Supplementary Data 3). These genes were all already known to be DD-associated, reflecting the fact that, in our burden analysis, 78% of the excess was in these known genes, meaning that only an additional ~1.4% of DDD probands (~157 probands) have a diagnostic variant in an X-linked gene not currently associated with DDs. Of these 23 genes, 19 passed Bonferroni correction in the combined analysis of both

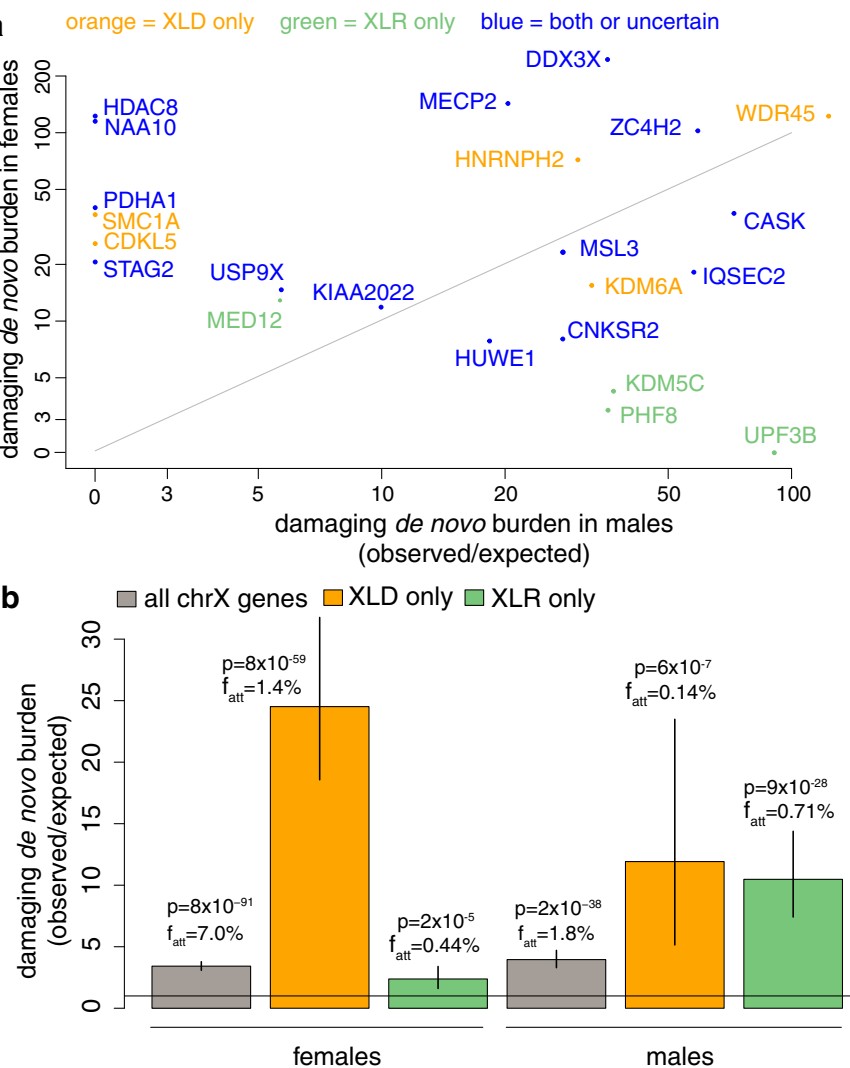

**Fig. 2 Sex-specific de novo burden analysis. a** Burden of damaging de novo mutations (DNMs) (protein truncating variants (PTVs) + missense/inframe) in females versus males, per gene. Shown are the 23 X-linked genes that passed multiple-testing correction. The text colour indicates whether the gene was classed in the consensus of genes from the Developmental Disorders Gene-to-Phenotype list (DDG2P) and Online Mendelian Inheritance in Man (OMIM) (see 'Methods') as X-linked dominant only (orange), X-linked recessive only (green) or both/uncertain (blue). P-values for the genes under different tests are shown in Supplementary Data 3. **b** Burden of damaging (PTV + missense/inframe) DNMs for males and females in the indicated gene sets. p: p-value from upper-tailed Poisson test. $f_{att}$: attributable fraction for DNMs in this gene set. The colored bars show the point estimates and error bars show 95% confidence intervals calculated as described in the 'Methods'.

sexes (our old method), one (*STAG2*) was only significant in the female-only test, and three were only significant when incorporating both de novo and inherited variants in males using the transmission and de novo association test (TADA)[27].

We observed that a subset of genes were significantly enriched for DNMs in females only (e.g. *HDAC8*, *NAA10*, *PDHA1*, *SMC1A*, *CDKL5*, *STAG2*), a subset only in males (*UPF3B*, *KDM5C*), and some in both sexes (*IQSEC2*, *CASK*, *WDR45*) (Fig. 2a). The patterns of enrichment we observed were largely consistent with the inheritance modes previously reported for these genes, with the exception of *MED12* which we discuss below. In principle, this kind of large-scale data analysis should allow us to explore modes of X-linked inheritance in a less biased way than previous small-scale case reports in the literature. We can see from burden analysis (Fig. 2b; positive predictive values shown in Supplementary Fig. 5) that genes annotated as XLD versus XLR clearly have different patterns of DNM enrichment, indicating that there is meaningful heterogeneity among X-linked genes. However, the fact that we still see enrichment of DNMs in

nominally XLR genes in females indicates that these classifications in the literature are not perfect.

*MED12* presents a good illustration of the challenges in trying to classify inheritance modes for X-linked genes. It had been previously reported to cause X-linked recessive FG, Lujan and Ohdo syndromes as well as non-syndromic intellectual disability[28–30], and DDG2P and OMIM class it as XLR. However, heterozygous females have previously been reported with mild and, in some cases, severe phenotypes, with the severity not obviously being correlated with the degree of skewed X-inactivation[31–34]. We observed eight damaging DNMs in females and one in a male. The phenotypes of our patients and those reported for other *MED12* patients in the literature are largely consistent. For five of the eight female *MED12* patients, the clinician reported that the contribution of the *MED12* mutation was 'uncertain', presumably partly because it is reported to be recessive in OMIM. Our results suggest that *MED12* should likely be reclassified as causing both X-linked dominant and X-linked recessive conditions, and be considered to cause a spectrum of

related disorders rather than distinct syndromes. Further work on the functional consequences of different *MED12* mutations and the degree of X-inactivation in female brain tissue would be required to understand why some mutations appear to be recessive and others dominant.

The *MED12* example above illustrates the power of these kinds of large-scale data analyses to identify patterns of sex-biased DNMs that are inconsistent with current classification of inheritance mode, but ideally we would like to be able to assign all genes to modes of inheritance with high confidence. Genes that exhibit a substantial female bias in observed DNMs (Fig. 2b) could be semi-dominant and lethal in males, or dominant (cause equivalent disease in males and heterozygous females). The low mutability of the maternally transmitted X chromosome in males results in a low expected number of DNMs in genes pathogenic in males. It may well be that we simply have not observed any males with DNMs in those genes by chance due to limited sample size. To distinguish these possibilities, we need to model the expected number of DNMs in females versus males, given the null mutation model, coverage, ploidy and sample size (see 'Methods'). Under this model we would expect ~78% of DNMs on the X chromosome to occur in females, although they only make up 43% of the trios used for the DNM enrichment tests. When we accounted for this, we identified two genes with a nominally significant female bias (Supplementary Table 3), *MECP2* (binomial $p = 0.02$; $24/25 = 96\%$ of DNMs observed in females) and *DDX3X* (binomial $p = 0.002$; $48/50 = 96\%$ of DNMs observed in females). This provides statistical evidence that both of these genes exhibit a semi-dominant mode of inheritance. These two genes were recently reported to show significant female bias for DNMs[35], although that work did not correct for the different mutation rates in males and females and hence overestimated the significance of the female bias. *MECP2* is already known to cause female-limited Rett syndrome, and males with verified PTV mutations have not been observed, although missense mutations have been reported in males with severe epileptic encephalopathy. In line with this, in both *MECP2* and *DDX3X*, the only DNMs in males in our cohort were missense or inframe. *MECP2* and *DDX3X* are among the genes with the most DNMs in our cohort, and therefore have the most power for assigning inheritance mode. Confident assignment of all X-linked genes to one or other inheritance mode would require a larger sample size than is available in this study.

**Role of polygenic background.** Clinicians recruiting patients to the DDD study were asked to indicate at recruitment whether they suspected the patient may have an X-linked cause. Male patients with suspected X-linked inheritance ($N = 271$) were enriched 2.7-fold for higher inherited X-linked coding burden (attributable fraction = 16.4% [6.4–27.4%] versus 6.1% [3.6–8.6%] for all males) (binomial $p = 2 \times 10^{-9}$). Based on recent work showing that common variants also contribute to risk of rare developmental disorders[36], we hypothesised that polygenic background could be contributing to the presence of multiple affected males in families, leading clinicians to incorrectly suspect X-linked inheritance. Using imputed genotype chip data on a subset of the cohort, we tested for a difference in polygenic scores for relevant traits between 216 males suspected to have X-linked inheritance versus 3439 who were not, having excluded those with a potentially diagnostic X-linked variant. Specifically, we assessed polygenic scores for educational attainment[37], intelligence[38], schizophrenia[39], and severe neurodevelopmental disorders (NDD) derived from our own GWAS[36]. None were significantly different after correcting for four tests (linear regression correcting for 10 genetic principal components; $p >$ 0.05/4; Supplementary Table 4). This analysis will require more powerful polygenic scores and a larger sample size to clarify the contribution of polygenic background to the clustering of affected individuals in families.

## Discussion

Here we analysed the burden of de novo and inherited rare coding variants on the X chromosome and estimated that these explain ~6% of both male and female probands in 11,044 families in the DDD study. We found that about three-quarters of this burden was in known DD genes. These proportions are similar to findings in other exome-sequencing studies of similar cohorts[40–42]. For example, in one of the largest comparable studies, 34/938 (3.6% [2.5–5.0%]) male probands and 29/728 (4.0% [2.7–5.7%]) female probands with neurological disorders had a diagnostic X-linked variant[42], versus 3.8% and 5.4%, respectively in DDD (inferred from burden analysis in known DD-associated genes). Importantly, our results show that, when rare monogenic causes in as-yet-undiscovered X-linked genes are also accounted for, they still cannot explain the male bias within our cohort. There is a common perception amongst clinicians that X-linked causes of DDs are much less likely in females than males (except for e.g. Rett Syndrome due to *MECP2* mutations)[43], but our burden analyses do not support this. Our results imply that the discovery of the remaining X-linked DD genes may allow us to diagnose another ~1.4% of our cohort. In contrast, we recently estimated that discovery of the remaining autosomal dominant DD genes would allow us to diagnose about another ~23% of the cohort with pathogenic DNMs[44]. This likely reflects the fact that it was easier to find X-linked DD genes than autosomal dominant DD genes in the linkage era.

We found that ~41% of the burden on the X chromosome in trio males was de novo, and this fraction was ~36% in XLR genes, higher than the ~18% expected under Haldane's theory. To our knowledge, this number has previously only been estimated for individual genes, and only based on likely diagnostic variants rather than in an unbiased burden analysis (e.g. refs. [45,46]). It is important to emphasise that the fractions we have estimated in DDD are likely higher than they would be in an unbiased sample of DD patients. DDD may be biased away from inherited X-linked causes due to our ascertainment strategy, which excluded patients who already had a genetic diagnosis. X-linked inherited causes may be easier to diagnose, since an X-linked inheritance pattern in a family reduces the search space. Additionally, the earlier GOLD study had already recruited several hundred of these families in the UK, so they may have been under-recruited to DDD. Reduced reproduction in heterozygous carrier females might also be contributing to the higher-than-expected contribution of DNMs in X-linked recessive genes; this could include increased pre-reproductive mortality in females with skewed X-inactivation as well as phenotypically normal mothers choosing not to have more children after having an affected son. We observed ~26% lower reproductive success (fertility ratio 0.742 [0.503–0.981]) in a small sample ($N = 13$) of carrier females in UK Biobank compared to non-carriers, but given the large confidence interval, this result should be treated with caution. Larger sizes of relatively unbiased population samples are needed to confirm this apparently decreased reproductive success, and to quantify its influence on the ratio of de novo to inherited pathogenic variants in XLR genes.

Our results have important diagnostic implications. We demonstrated that, while the vast majority of observed de novo and inherited PTVs in males in known X-linked DD genes are pathogenic, inherited missense variants in these genes have only a low PPV (Fig. 1c; Supplementary Fig. 4). This implies that it is

challenging to accurately diagnose males with rare inherited missense variants in X-linked DD genes, particularly since the high recurrence risk for such variants presents a legitimate concern and thus may increase motivation for clinicians to take action. Additional simple manoeuvres such as testing the apparently unaffected males in the family for an inherited missense variant identified in an index male can be very helpful in excluding causality but are rarely deployed outside of clinical genetics services. On the other hand, our comparison with ClinVar (Fig. 1b) suggests that, in fact, current clinical interpretation is likely highly conservative and missing a large fraction of pathogenic missense variants in X-linked DD genes in males, since PTVs are relatively enriched in the ClinVar likely pathogenic/pathogenic variants compared to our burden analysis. Indeed, assuming that all diagnostic PTVs are being identified, we estimate from this analysis that ~58% of diagnostic missense/inframe variants in known X-linked DDG2P genes are not being classed as pathogenic. Incorporating CADD and MPC scores during variant interpretation may improve specificity in clinical diagnosis, but reduce sensitivity (Fig. 1c). For example, considering all inherited missense variants absent from gnomAD in DDG2P genes, 13% [4–22%] of these will be truly pathogenic, versus 53% [37–65%] of those with MPC score > 2, but applying the latter filter may lose about a third of diagnoses.

We developed an improved strategy for finding X-linked Mendelian disease genes that considers several different inheritance modes and incorporates both de novo and inherited variation in males, incorporating and building on the TADA method[27]. We showed that this strategy identified 21% (23 versus 19) more X-linked DD-associated genes at genome-wide significance than our previous approach. However, all genes that passed genome-wide significance were already known, reflecting low power which is due to the fact that only ~1.4% of the cohort has an X-linked diagnosis in an as-yet-undiscovered gene. Although our approach was designed to distinguish XLD from XLR genes, it is clear that more data will be needed from larger cohorts for a data-based classification of inheritance modes. Furthermore, information in X-inactivation in females may help us to interpret inheritance modes, since this does contribute to penetrance in females, although accessibility of the relevant tissue is likely to be a challenge here. Larger sample sizes may reveal more X-linked genes with a sex bias other than just DDX3X and MECP2[35] which appear to be semi-dominant; we emphasise that, in testing for these, it will be important to account for the differential rates of DNMs on the X chromosome in the two sexes, otherwise the degree of female bias will be over-estimated (see 'Methods').

Our study has several limitations. Firstly, there are wide confidence intervals around several of the key parameters we have estimated (attributable fraction, fraction of de novo versus inherited causes, positive predictive value, etc.), despite our relatively large sample size. An even larger sample size would increase the precision of these estimates. Secondly, there may be ascertainment bias in the DDD study away from very recognisable disorders and families with a clear XLR inheritance pattern. This means our estimates of important parameters will not necessarily hold for other cohorts. Large cohorts in which exome-sequencing has been applied as a first-line test will allow less biased estimates. Finally, our attempts at new gene discovery and classification of inheritance modes were limited not only by sample size, but also by lack of data on X-inactivation in females and the phenotypes of carrier mothers. Furthermore, we anticipate that extra data will reinforce that many X-linked genes can show both dominant and recessive inheritance depending on the severity of the variant, including, for some genes, different phenotypic features associated with hemizygous versus heterozygous

variants (e.g. NAA10[47]). Hence, future gene discovery efforts will need to develop unbiased ways of discriminating between different disorders caused by mutations in the same gene on the basis of phenotype or functional evidence.

It is notable that within our cohort, less than a quarter of males who were suspected by clinicians to have an X-linked condition were inferred to have a pathogenic rare or de novo X-linked coding variant in any gene. This may imply that there are other factors contributing to the recurrence of a DD in multiple males from the same family. Under the hypothesis that males have a lower liability threshold than females for NDDs, it seems plausible that an enrichment of multiple deleterious variants across the frequency spectrum might push multiple males but not females in a family over this threshold, creating the appearance of X-linked Mendelian inheritance. We did not see a difference in polygenic scores (with MAF > 5% variants) for relevant traits between males without likely diagnostic X-linked variants who were suspected to have an X-linked disorder versus those who were not. However, given that existing polygenic scores for intelligence explain only ~4–5% of variance in IQ in out-of-sample prediction[48–50], we anticipate that we may see a difference with a larger sample size and more informative polygenic scores, potentially including rarer variants that have been shown to explain a substantial proportion of variance in intelligence[51].

In conclusion, our work shows that monogenic causes on the X chromosome cannot account for the male bias in developmental disorders. Analyses of variants across the full frequency spectrum in large cohorts may reveal a contribution of more common variants to the sex bias. This work provides a robust statistical framework for analyses of the X chromosome in large Mendelian disease cohorts, which will aid in future gene discovery and inform improvements in clinical practice.

## Methods

**Family recruitment**. Individuals with severe, undiagnosed developmental disorders were recruited to the DDD project by 24 clinical genetics centres within the United Kingdom National Health Service and the Republic of Ireland. They had to have at least one of the following phenotypes:

1. Neurodevelopmental disorder—for example, developmental delay and/or learning disability (of a level requiring or likely to require a statement of special educational needs), epileptic encephalopathy or cerebral palsy.

2. Congenital anomalies—multiple congenital anomalies (two or more major anomalies) or a single major anomaly together with a neurodevelopmental disorder, aberrant growth, dysmorphic features or unusual behaviour.

3. Abnormal growth parameters (height, weight, head circumference (OFC))—two or more parameters >3 SD above or below the mean or a single parameter >4 SD above or below the mean (except for obesity where the threshold for isolated obesity is >4.5 SD together with a strong suspicion of a genetic aetiology).

4. Unusual behavioural phenotype in conjunction with one or more of the above features or extreme behavioural phenotype strongly suspected to have a genetic basis (including classical autism).

5. Genetic disorder of significant impact for which the molecular basis is currently unknown with: (i) several affected family members or (ii) one other affected family member with a rare, consistent and distinctive phenotype or (iii) a single case that is associated with a severe phenotype.

Families gave informed consent to participate, and the study was approved by the UK Research Ethics Committee (10/H0305/83, granted by the Cambridge South Research Ethics Committee and GEN/284/12, granted by the Republic of Ireland Research Ethics Committee). DNA was collected from saliva samples obtained from the probands and their parents, and from blood obtained from the probands. The individuals analysed in this paper include those analysed in the previous publications[12,44,52,54,55].

**Clinical features**. The patients were systematically phenotyped using Human Phenotype Ontology (HPO) terms[56], and growth measurements, developmental milestones, family history (including whether X-linked inheritance was suspected) etc. were collected within DECIPHER[57]. For the summary of phenotypes and comparison between the sexes (Supplementary Fig. 1, Supplementary Data 1), we followed the procedure in ref. [55] when counting organ systems affected, to avoid double-counting HPO terms that fall under multiple organ systems. For the comparison of age at walking and talking in Supplementary Table 1, we excluded probands who had not yet reached these milestones; the sample sizes are shown in

the table. We used linear regression correcting for age at assessment to compare quantitative phenotypes between sexes, and logistic regression correcting for age at assessment to compare the frequency of binary phenotypes between sexes.

**Exome sequencing, variant annotation and variant quality control.** We carried out Illumina exome sequencing using the Agilent v3 or v5 baits[12]. Mapping of short-read sequences for each sequencing lanelet was carried out using the Burrows-Wheeler Aligner using both the aln algorithm (BWA version 0.5.10) and the bwa-mem algorithm (BWA version 0.7.12), with the GRCh37 1000 Genomes Project phase 2 reference (also known as hs37d5). Sample-level BAM improvement was carried out using the Picard Markduplicates (versions 1.98 and 1.114) and Genome Analysis Toolkit IndelRealigner (GATK version 3.1.1 and version 3.5.0), which performs realignment of reads around known and discovered indels (insertions and deletions). Single-nucleotide variants (SNVs) and indels were called using the GATK HaplotypeCaller, CombineGVCFs and GenotypeGVCFs (GATK version 3.5.0). Bcftools (version 1.8-30-gb717d08) and custom Perl (version 5) scripts were used to filter the variants for the case/control analysis.

Variants were annotated with Ensembl Variant Effect Predictor[58] based on Ensembl gene build 88, using the LOFTEE plugin. We analysed three categories of variant based on the predicted consequence: (1) protein-truncating variants (PTVs) classed as 'high confidence' loss-of-function variants by LOFTEE (including the annotations splice donor, splice acceptor, stop gained, frameshift, initiator codon and conserved exon terminus variant); (2) missense variants and inframe indels; (3) synonymous variants. We assigned each variant the worst consequence across all the transcripts for a gene. Missense variants were annotated with CADD v1.3[21] and MPC[22] scores. All variants were annotated with MAF data from four different populations of the 1000 Genomes Project[59] (American, Asian, African and European), two populations from the NHLBI GO Exome Sequencing Project (European Americans and African Americans) and six populations from the Genome Aggregation Consortium (gnomAD) release 2.0.2 (African, East Asian, non-Finnish European, Finnish, South Asian, Latino,), and internal allele frequencies from the European and South Asian unaffected DDD parents.

For the case-control analysis of chrX in males, we used the following filters:

- Genotypes were set to missing if they had genotype quality (GQ) < 20, depth (DP) < 7, or were called as heterozygous.
- Variants were removed if they met any of the following criteria:

  - were in the pseudoautosomal regions (chrX:60001-2699520 and chrX:154931044-155260560 in GCh37)
  - had a strand bias p-value < 0.001
  - had >50% missing calls (after the genotype-level filtering) within the samples that underwent Agilent SureSelect Human All Exon V5 capture or within those that underwent Agilent SureSelect Human All Exon V3 capture
  - had minor allele frequency (MAF) > 0.001 in any gnomAD population or in the unaffected European or South Asian parents from DDD. For calculating the PPV for lower MAF variants (MAF < 0.0005, 0.0001, 0.00005), we only considered the gnomAD POPMAX and the frequency in European DDD parents, since the set of South Asian DDD parents in DDD was too small.
  - had any hemizygotes in gnomAD

For calling DNMs on chrX, we ran DeNovoGear[12,60], but with a different set of hard filters to account for the lower coverage in males and to maximise sensitivity and specificity. We examined all candidate DNMs in males and a large subset of those in females manually in IGV, and used this to settle on the following set of filters:

- We removed DNMs in the pseudoautosomal regions.
- The variant had to be called heterozygous or (for males) hemizygous in the child in the original GATK calls, and called homozygous reference in the parents.
- For male probands, we required the following: in the child, alternate allele depth >2 and overall depth > 2; in the mother, depth >5.
- For female probands, we required the following: in the child, alternate allele depth >2, overall depth >7; in the mother, depth >5; in the father, depth >1.
- For single nucleotide variants, we required $p > 10^{-3}$ on a Fisher's exact test for strand bias, pooling across trios (or mother-child pairs, for male probands) where a DNM was called at the same site by DeNovoGear.
- For female probands, we removed indels <5 bp if they had variant allele fraction <0.3 or MAF > 0, since these were vastly over-represented and seemed to be a common error mode.
- We did a two-sided binomial test on the number of alternate reads at the candidate site in mothers, assuming the proportion of these should be 0.5 if heterozygous, and then discarded sites where the p-value from this test (called $p_{het}$) was >0.01, since these indicated that the mother was likely to be truly heterozygous and not mosaic.
- We did a binomial test to evaluate whether the fraction of alternate reads was greater than the expected error rate of 0.2% ($p_{error}$), and then flagged variants as mosaic if they had lower-tailed $p_{het} < 0.01$ and upper-tailed $p_{error} < 0.01$.

We then removed variants in segmental duplications if both the child and at least one parent (mother for female probands) were flagged as mosaic.

- We set a cutoff for the posterior probability of being a DNM from DeNovoGear to $pp_{DNM} > 0.00247679$. This value was chosen because the observed number of synonymous DNMs in females that passed this $pp_{DNM}$ cutoff, as well as the hard filters above, was very close to the expected number. calculated using null mutation rate determined as described below. We chose to calibrate the $pp_{DNM}$ threshold in females since the numbers in males were very small.

We subsequently removed DNMs that had MAF > 0.001 in any gnomAD population or in the unaffected European or South Asian parents from DDD, or that had any hemizygotes in gnomAD.

**Calculation of expected mutation rates on chrX.** We estimated the expected number of DNMs per gene in different functional classes using the method in ref. [10]. We adjusted for the reduced sensitivity to detect DNMs due to limited coverage following the method in ref. [61], with some minor adaptations. For this, we first calculated the median depth per exon in 250 samples on the Agilent V5 capture, and then took the mean of these. To determine the depth-uncorrected expected number of variants per exon, we took the exons with mean median depth ≥30 and regressed the number of rare (MAF < 0.001) synonymous variants on the probability of a synonymous mutation. For males and females separately, we plotted the ratio of observed to depth-uncorrected expected synonymous variants against the depth in bins of 2× (for up to 40× in males and 80× in females) and fitted a logarithmic curve. We then used this formula to predict the depth-corrected expected number of variants for all exons:

$$\text{depth-adjusted expected count} =$$
$$\begin{cases} 0 \text{ if depth} < 1 \\ \text{expected count} \times (0.2778 \ln(\text{depth}) + 0.0279) \text{ if } 1 < \text{depth} < 30 \text{ in males} \\ \text{expected count} \times (0.2464 \ln(\text{depth}) + 0.035) \text{ if } 1 < \text{depth} < 50 \text{ in females} \\ \text{expected count if depth} > 30 \text{ in males or} > 50 \text{ in females} \end{cases} \quad (1)$$

In calculating the expected counts of DNMs in the non-pseudoautosomal regions of chrX in males and females, we followed the method previously described in[53] to account for the different inheritance pattern of this chromosome and the different mutation rates in the male and female germline. Specifically, we determined the scaling factors as follows:

$$f_{females} = t_{F>F} + t_{M>F} = n_{female}\lambda_{female} + n_{female}\lambda_{male} \quad (2)$$

$$f_{males} = t_{F>M} = n_{male}\lambda_{female} \quad (3)$$

where $t_{F>F}$ and $t_{M>F}$ are the number of transmissions from females and males to female probands respectively, $t_{F>M}$ is the number of transmissions from females to male probands, $n_{female}$ and $n_{male}$ are the numbers of female and male probands, and $\lambda_{female}$ and $\lambda_{male}$ are adjustment factors to account for the higher mutation rate in males, given by:

$$\lambda_{female} = \frac{2}{1+\alpha} \text{ and } \lambda_{male} = \frac{2}{1+\frac{1}{\alpha}} \quad (4)$$

where $\alpha = 3.4$ is the ratio of the mutation rate in fathers to mothers in DDD, determined using 199 phased DNMs[53]. These scaling factors $f_{females}$ and $f_{males}$ were multiplied by the depth-adjusted mutation rates to determine the expected number of DNMs as $n_{female}\mu_{female}$ and $n_{male}\mu_{male}$.

**Sample quality control.** We removed probands with sex chromosome aneuploidies ($N = 47$) or whose chromosomal sex did not match the sex recorded by the clinician (N = 49). We also removed six probands with an implausibly high number of de novo calls, likely to be spurious, and 330 samples with >20% missing genotypes on chrX after the genotype QC described above. We used KING[62] to estimate relatedness between individuals, applying it to variants with MAF > 0.01 with <5% missingness after genotype QC. Then, for unaffected fathers (who were used as controls) and for probands separately, we removed one person from each pair of individuals inferred to be third-degree relatives or closer (doing this in such a way as to minimise the number of individuals removed); this led to the removal of 586 male probands, 211 female probands and 108 unaffected fathers. In total, 1311 unique individuals were removed. The final analysis after QC was conducted on 7136 male probands (5138 in trios), 3908 female probands in trios, and 8551 unaffected fathers.

**Burden analysis.** We conducted sex-specific burden analyses to test for an enrichment of certain classes of X-linked variants in probands and to estimate the fraction of probands attributable to such variants.

1. In males, we conducted a case/control analysis comparing male probands to the unaffected fathers. This incorporates both maternally inherited variants and DNMs that passed the genotype and variant filtering for the case/control analysis (but did not necessarily pass the filtering of mutations called

by DeNovoGear). In the case-control analysis, we calculated the rate of a particular class of variant per person in the unaffected fathers and used this to calculate the expected number of variants in the male probands by simply multiplying this rate by the number of probands.

2. In both females and males, we conducted a DNMs enrichment analysis comparing the observed number of DNMs to the expected number, as calculated above.

For both burden analyses, we restricted to variants with maximum MAF < 0.001 and with no hemizygotes in gnomAD, to try to restrict to the most damaging subset of variants. We also examined synonymous, PTV and missense/inframe variants (the latter filtered in various ways) separately. Synonymous variants were used as a control: we confirmed that the number of observed synonymous variants was not significantly different from expectation, to ensure that the test was well-calibrated.

We tested for enrichment assuming a Poisson distribution using an upper-tailed test. The metrics reported in the main text were calculated as follows:

$$burden = \frac{\# \ observed \ variants}{\# \ expected \ variants} \quad (5)$$

$$attributable \ fraction = \frac{excess}{\#probands} = \frac{\#observed \ variants - \#expected \ variants}{\#probands} \quad (6)$$

$$PPV = \frac{excess}{\#observed \ variants} = \frac{\#observed \ variants - \#expected \ variants}{\#observed \ variants} \quad (7)$$

For DNMs, we treated the observed number as a fixed quantity, calculated 95% confidence intervals on the number of expected DNMs using the poisson.test() function in R, and then substituted these into the above formulae to calculate confidence intervals on those metrics. For the case/control analysis in males, the expected rate of variants is calculated based on the number observed in fathers, so is a random variable. Thus, we used the moveci() function in R to calculate confidence intervals on the difference or ratio of two Poisson rates, and substituted these back into the above formula as appropriate. Specifically, the confidence interval for each metric was calculated as follows:

- burden: moveci ($V_{probands}$, $N_{probands}$, $V_{fathers}$, $N_{fathers}$, distrib = 'poi', contrast = 'RR') (i.e. a rate ratio) where $N$ is the number of individuals and $V$ is the number of observed variants.
- attributable fraction: moveci ($V_{probands}$, $N_{probands}$, $V_{fathers}$, $N_{fathers}$, distrib = 'poi', contrast = 'RD') (i.e. a difference in rates)
- PPV: 1-moveci ($V_{fathers}$, $N_{fathers}$, $V_{probands}$, $N_{probands}$, distrib = 'poi', contrast = 'RR')

To calculate a confidence interval on the fraction of pathogenic variants that were in known genes and the fraction of pathogenic variants in XLR genes in male trio probands that was de novo, we bootstrapped probands. Specifically:

- For females, we bootstrapped probands 1000 times, each time recalculating the excess number (excess = #observed − #expected) of DNMs in all genes and in known X-linked DD genes, then determined the 2.5th and 97.5th percentile of the ratio $\frac{excess_{DNM,known \ genes}}{excess_{DNM,all \ genes}}$ across the 1000 iterations (Supplementary Fig. 2B).
- For males, we bootstrapped probands and fathers each separately 1000 times, each time recalculating the excess number of variants in probands in all genes and in known X-linked DD genes, then determined the 2.5th and 97.5th percentile of the ratio $\frac{excess_{vars,known \ genes}}{excess_{vars,all \ genes}}$ across the 1000 iterations (Supplementary Fig. 2A). We repeated this procedure with just the male trio probands, and used the same 1000 sets of bootstrapped trios to also calculate $excess_{vars,all \ genes}$, $excess_{vars,XLR \ genes}$, $excess_{DNM,all \ genes}$ and $excess_{DNM,XLR \ genes}$, then determined the 5th and 95th percentile of the ratios $\frac{excess_{DNM,all \ genes}}{excess_{vars,all \ genes}}$ (Supplementary Fig. 2C) and $\frac{excess_{DNM,known \ genes}}{excess_{vars,known \ genes}}$ (Supplementary Fig. 2D). In Supplementary Fig. 2DA, C, the 97.5th percentile of the ratio was greater than 1, so in the text we report the upper bound of those confidence intervals to be 100%.

**Per-gene enrichment tests**. For each gene, we tested for a significant burden of DNMs using a Poisson test, calculating the expected number of DNMs using the expected mutation rate obtained as described above[53]. For males and females combined (the old test) and for females alone, we tested PTVs alone and PTVs and missense/inframe variants combined, then took the lowest p-value for each proband set.

We also applied the transmission and de novo association test (TADA)[27] to each gene in males to test for enrichment of de novo and inherited variants combined. The inherited counts were determined from the male probands and their fathers, respectively. We removed from these counts the de novos that passed our DeNovoGear filtering, since these were counted already in the TADA de novo mutation model; we also tried removing from the inherited counts variants that did not pass the DeNovoGear filtering but which had 0 alternate reads in mothers, and counting these as DNMs, but in practice this made little difference to the results. Since there are ~15% of genes on chrX already implicated in DDs (in the DDG2P list) but it

was unclear how the case ascertainment in DDD might have created biases against these, we tried varying $\pi$, the prior on the fraction of risk genes, from 0.05 to 0.25. Other prior parameters were calculated accordingly, following the procedure described in the TADA user guide (http://www.compgen.pitt.edu/TADA/TADA_guide.html). These are shown in Supplementary Table 5. For all runs, as parameters in the prior for the allele frequencies, we set $v = 100$ and $\rho = 0.618$ for PTVs, and $v = 100$ and $\rho = 11.749$ for missense/inframe variants. We ran TADA separately on PTVs alone and on PTVs and missense/inframe variants combined.

We calculated $q$-values using the Bayesian.FDR() function in TADA, and $p$-values using a sampling approach via the TADAnull() and bayesFactor.pvalue() functions.

Under the old testing strategy (PTVs alone and PTVs+missense/inframe variants combined for males and females combined), we corrected for $2 \times 19,685$ genes, giving a genome-wide significance threshold of $p < 0.05/2 \times 19,685 = 1.27 \times 10^{-6}$. Under our improved testing strategy, overall we applied six tests to each of 804 X-linked genes: (A) PTVs alone and (B) PTVs+missense/inframe combined for each of (1) DNMs in females alone (Poisson), (2) DNMs in males and females combined (Poisson), and (3) DNMs and inherited variants in males (TADA). Since we would typically apply two separate DNM enrichment tests to each autosomal gene (PTVs alone and PTVs+missense/inframe variants), in total, we corrected for $(6 \times 804 + 2 \times (19,685 - 804))$ tests, giving a genome-wide significance threshold of $p < 0.05/40780 = 1.17 \times 10^{-6}$.

**Testing for sex bias in DNMs per gene**. For each gene, we calculated the fraction of expected PTV+missense/inframe DNMs that should be in males as:

$$expected \ fraction \ of \ DNMs \ in \ males = \\ n_{male}\mu_{male}/(n_{male}\mu_{male} + n_{female}\mu_{female}) \quad (8)$$

We used the mutation rates calculated as described above accounting for coverage and ploidy. We then compared the fraction of observed DNMs that were in males to this expected fraction using a lower-tailed binomial test (under the hypothesis that the gene would be depleted for DNMs in males due to lethality).

**Gene list definitions**. To define the list of known X-linked DDG2P genes, we took the intersection of confirmed or probable DDG2P genes on the X chromosome and OMIM genes with a disease annotation. To define 'X-linked recessive' (called 'hemizygous' in DDG2P) and 'X-linked dominant' genes, we compared the inheritance annotations between DDG2P and OMIM and took the consensus. Hence, 'X-linked recessive' genes were those annotated only as hemizygous in DDG2P and only as X-linked recessive in OMIM, and similarly for X-linked dominant genes. There were 12 genes classified as exclusively X-linked dominant and 63 as exclusively X-linked recessive. Genes annotated as both X-linked dominant and X-linked recessive, or annotated simply as 'X-linked' in either DDG2P and OMIM, have been coloured in blue in Fig. 2, and excluded from analysis of X-linked recessive genes and from Supplementary Figs. 2D, 4 and 5.

**Investigating the de novo versus inherited contribution in X-linked recessive genes**. Under Haldane's theory, the fraction of male X-linked recessive cases due to DNMs, $p_{\text{de novo}}$, should be:

$$p_{de \ novo,Haldane} = \frac{m\mu}{2\mu + \nu} \quad (9)$$

where $m$ is the reproductive loss in affected males, $\mu$ is the mutation rate in eggs and $\nu$ in the mutation rate in sperm. If we assume $m = 1$ and that $\nu = 3.5\mu$ (based on a previous estimate[24]), we should expect $p_{\text{de novo}}$ to be about $\frac{1}{5.5} \simeq 18.2\%$.

We focused on X-linked recessive genes from DDG2P that were not also annotated as X-linked dominant, and included only PTV and missense SNVs because the male to female mutation rate ratio was calculated for SNVs. We estimated (based on the excess=observed-expected) that 68.5 [31.2–102.9] male trio cases could be attributed to pathogenic variants in these genes, of which 21.8 [15.5–24.8] were de novo. Assuming the data approximate a binomial distribution, we tested the consistency with Haldane's theory using a two-sample test for equality of proportions; specifically, we used prop.test(22,69, $p = 1/5.5$) in R. We also assessed consistency with Haldane's theory through bootstrapping as described in the "Burden analysis" section above.

Sherman et al. modified Haldane's theory to account for reduced reproductive fitness in carrier females[26]. Under their theory, $p_{\text{de novo}}$ should be:

$$p_{de \ novo,Sherman} = \frac{\mu(2f + m - mf)}{2\mu + \nu - \nu f} \quad (10)$$

where $f$ is the reproductive loss in carrier females. We again assume $m = 1$ and $\nu = 3.5\mu$. Any value of $p_{\text{de novo}}$ between 0.22 and 0.44 should be consistent (p-value > 0.05) with the excess values we observed. This corresponds to a value for $f$ between ~12% and ~56%. We next turned to the UK Biobank exome data to estimate $f$ directly.

**UK Biobank analysis**. We downloaded variant calls for all 49,959 samples subjected to WES as part of the UK Biobank study[63]. We next annotated all variants with Variant Effect Predictor (v97)[58], extracted variants in exclusively X-linked recessive genes, and retained only the most severe consequence in canonical

transcripts. We note that none of these X-linked recessive genes are affected by the recently-reported problem with this UK Biobank exome data release which is related to mapping errors[64].

We then removed PTV and missense variants with CADD ≤ 25[65], missense variants with an MPC[22] score ≤2, low-confidence PTV variants using the LOFTEE plugin for VEP[66], and all variants with a gnomAD[19] non-Finish European minimum allele frequency ≥0.001. Only UK Biobank variants with a UK Biobank allele frequency ≤$1 \times 10^{-3}$ were retained for downstream analysis. To further ensure that we did not include any deleterious variants of potentially incomplete penetrance, we also filtered any PTV and missense variants which were also found in any male individuals in UK Biobank. We then removed all related, non-white British ancestry individuals and all males, leaving a final total of 18,632 female samples. PTV, missense, and synonymous variants passing the above criteria were then counted for each remaining individual (Supplementary Fig. 4A). All 13 likely PTV variants in X-linked recessive genes we identified were confirmed via manual inspection using the Integrative Genomics Viewer[67] (Supplementary Table 4).

To determine the total number of live births for each female in UK Biobank, we downloaded field 2734. To determine fluid intelligence scores, we downloaded field 20016. Only data obtained via in-centre testing were retained for further analysis. Age and pre-computed ancestry PCs[68] were obtained from UK Biobank fields 21022 and 22009, respectively. To independently determine the effect of PTV, missense, or synonymous variants on each phenotype, we used a simple linear model (via the glm function in R with family "gaussian") in the form:

$$phenotype \sim N_{VAR} + age + age^2 + PC1+....+PC10 \quad (11)$$

where phenotype is either number of live births or normalised fluid intelligence, and $N_{VAR}$ is one of total PTV, missense, or synonymous variants in each individual in all confirmed X-linked recessive DD genes (Supplementary Fig. 4B, C). To determine the ratio of number of live births between X-linked recessive PTV carrier and non-carrier females, corrected for age and ancestry PCs, we used the function ttestratio from the R package mratios v1.4.0 with default settings.

**Polygenic score analysis**. We restricted the analysis to 4168 male probands with a neurodevelopmental disorder who had been genotyped on the CoreExome array, had European ancestry and passed our quality control in[36]. We excluded 513 males who had an X-linked variant in a DDG2P gene reported to DECIPHER that had not yet been classed as 'benign' or 'likely benign' by clinicians (but note that many of these variants reported in DECIPHER had not yet been evaluated by clinicians and are likely to be deemed benign eventually). Polygenic scores for educational attainment[37], intelligence[38] and schizophrenia[39] were calculated using the pruning and thresholding method (see parameters in Extended Data Table 2 of ref. [36]). For calculating the NDD polygenic score, we repeated the GWAS in the same dataset described in ref. [36] but without sex as a covariate, then used $p < 1$ and $r^2 < 0.1$ when pruning the SNPs. For the comparisons of polygenic scores between males who were versus were not suspected by clinicians to have an X-linked diagnosis, we ran a linear regression of polygenic score on suspected group status plus 10 genetic principal components.

**Reporting summary**. Further information on research design is available in the Nature Research Reporting Summary linked to this article.

## Data availability

The full variant call files used in this study are accessible in the European Genome-Phenome Archive as dataset EGAD00001004389, and a file of phenotypic and family descriptions under EGAD00001004388. Both of these are under managed access to ensure that the work proposed by the researchers is allowed under the study's ethical approval. The de novo mutations used in the analysis are in Supplementary Data 2. Databases used in this study: Online Mendelian Inheritance in Man https://omim.org/; ClinVar https://www.ncbi.nlm.nih.gov/clinvar/; Developmental Disorder Gene-to-Phenotype list https://www.ebi.ac.uk/gene2phenotype/downloads; GRCh37 1000 Genomes Project phase 2 reference (hs37d5) https://www.ncbi.nlm.nih.gov/assembly/GCF_000001405.13/.

## Code availability

Code used to implement the analysis in this paper is available at https://github.com/hilarymartin/DDD_chrX[69].

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

## Acknowledgements

We thank the DDD families for participating, the DDD clinicians for recruiting patients, the Sanger Sample Management and Sequencing pipelines teams for generating the data, the Sanger Human Genome Informatics team for helping to process the exome data, and Nicola Whiffin for helpful comments on the manuscript. The study was approved by the UK Research Ethics Committee (10/H0305/83, granted by the Cambridge South Research Ethics Committee and GEN/284/12, granted by the Republic of Ireland Research Ethics Committee). This work has been conducted using the UK Biobank Resource under Application Number 44165. The DDD study presents independent research commissioned by the Health Innovation Challenge Fund (grant HICF-1009-003), a parallel funding partnership between the Wellcome Trust and the UK Department of Health, and the Wellcome Trust Sanger Institute (grant WT098051). The views expressed in this publication are those of the author(s) and not necessarily those of the Wellcome Trust or the UK Department of Health. The study has UK Research Ethics Committee approval (10/H0305/83, granted by the Cambridge South Research Ethics Committee and GEN/284/12, granted by the Republic of Ireland Research Ethics Committee). The research team acknowledges the support of the National Institutes for Health Research, through the Comprehensive Clinical Research Network. This study makes use of DECIPHER (http://decipher.sanger.ac.uk), which is funded by the Wellcome Trust.

## Author contributions

H.C.M. analysed the DDD data and drafted the manuscript together with M.E.H. who conceived and supervised the study. J.K. and K.E.S. assisted with the filtering of DNMs in the DDD study and modelling of mutation rates. A.S., N.A. and J.M. contributed to analyses of the DDD exome data. R.Y.E. and G.G. conducted quality control on the DDD exome data. E.J.G. and M.D.C.N. analysed the UK Biobank exome data. A.L.T.T. examined clinical features of patients. M.E.K.N processed the chip genotype data and constructed polygenic scores. C.F.W., D.R.F. and H.V.F. provided clinical and analytical supervision.

## Competing interests

M.E.H. is a co-founder of, consultant to, and holds shares in Congenica Ltd., a genetics diagnostics company. J.F.M. is an employee of Illumina Inc. A.L.T.T. is an employee of Genomics England. The other authors declare no competing interests.

## Additional information

## Deciphering Developmental Disorders Study

Silvia Borras[11], Caroline Clark[11], John Dean[11], Zosia Miedzybrodzka[11], Alison Ross[11], Stephen Tennant[11], Tabib Dabir[12], Deirdre Donnelly[12], Mervyn Humphreys[12], Alex Magee[12], Vivienne McConnell[12], Shane McKee[12], Susan McNerlan[12], Patrick J. Morrison[12], Gillian Rea[12], Fiona Stewart[12], Trevor Cole[13], Nicola Cooper[13], Lisa Cooper-Charles[13], Helen Cox[13], Lily Islam[13], Joanna Jarvis[13], Rebecca Keelagher[13], Derek Lim[13], Dominic McMullan[13], Jenny Morton[13], Swati Naik[13], Mary O'Driscoll[13], Kai-Ren Ong[13], Deborah Osio[13], Nicola Ragge[13], Sarah Turton[13], Julie Vogt[13], Denise Williams[13], Simon Bodek[14], Alan Donaldson[14], Alison Hills[14], Karen Low[14], Ruth Newbury-Ecob[14], Andrew M. Norman[14], Eileen Roberts[14], Ingrid Scurr[14], Sarah Smithson[14], Madeleine Tooley[14], Steve Abbs[4], Ruth Armstrong[4], Carolyn Dunn[4], Simon Holden[4], Soo-Mi Park[4], Joan Paterson[4], Lucy Raymond[4], Evan Reid[4], Richard Sandford[4], Ingrid Simonic[4], Marc Tischkowitz[4], Geoff Woods[4], Lisa Bradley[15], Joanne Comerford[15], Andrew Green[15], Sally Lynch[15], Shirley McQuaid[15], Brendan Mullaney[15], Jonathan Berg[16], David Goudie[16], Eleni Mavrak[16], Joanne McLean[16], Catherine McWilliam[16], Eleanor Reavey[16], Tara Azam[10], Elaine Cleary[10], Andrew Jackson[10], Wayne Lam[10], Anne Lampe[10], David Moore[10], Mary Porteous[10], Emma Baple[17], Júlia Baptista[17], Carole Brewer[17], Bruce Castle[17], Emma Kivuva[17], Martina Owens[17], Julia Rankin[17], Charles Shaw-Smith[17], Claire Turner[17], Peter Turnpenny[17], Carolyn Tysoe[17], Therese Bradley[18], Rosemarie Davidson[18], Carol Gardiner[18], Shelagh Joss[18], Esther Kinning[18], Cheryl Longman[18], Ruth McGowan[18], Victoria Murday[18], Daniela Pilz[18], Edward Tobias[18], Margo Whiteford[18], Nicola Williams[18], Angela Barnicoat[19], Emma Clement[19], Francesca Faravelli[19], Jane Hurst[19], Lucy Jenkins[19], Wendy Jones[19], V.K.Ajith Kumar[19], Melissa Lees[19], Sam Loughlin[19], Alison Male[19], Deborah Morrogh[19], Elisabeth Rosser[19], Richard Scott[19], Louise Wilson[19], Ana Beleza[20], Charu Deshpande[20], Frances Flinter[20], Muriel Holder[20], Melita Irving[20], Louise Izatt[20], Dragana Josifova[20], Shehla Mohammed[20], Aneta Molenda[20], Leema Robert[20], Wendy Roworth[20], Deborah Ruddy[20], Mina Ryten[20], Shu Yau[20], Christopher Bennett[21], Moira Blyth[21], Jennifer Campbell[21], Andrea Coates[21], Angus Dobbie[21], Sarah Hewitt[21], Emma Hobson[21], Eilidh Jackson[21], Rosalyn Jewell[21], Alison Kraus[21], Katrina Prescott[21], Eamonn Sheridan[21], Jenny Thomson[21], Kirsty Bradshaw[22], Abhijit Dixit[22], Jacqueline Eason[22], Rebecca Haines[22], Rachel Harrison[22], Stacey Mutch[22], Ajoy Sarkar[22], Claire Searle[22], Nora Shannon[22], Abid Sharif[22], Mohnish Suri[22], Pradeep Vasudevan[23], Natalie Canham[24], Ian Ellis[24], Lynn Greenhalgh[24], Emma Howard[24], Victoria Stinton[24], Andrew Swale[24], Astrid Weber[24], Siddharth Banka[25], Catherine Breen[25], Tracy Briggs[25], Emma Burkitt-Wright[25], Kate Chandler[25], Jill Clayton-Smith[25], Dian Donnai[25], Sofia Douzgou[25], Lorraine Gaunt, Elizabeth Jones[25], Bronwyn Kerr[25], Claire Langley[25], Kay Metcalfe[25], Audrey Smith[25], Ronnie Wright[25], David Bourn[26], John Burn[26], Richard Fisher[26], Steve Hellens[26], Alex Henderson[26], Tara Montgomery[26], Miranda Splitt[26], Volker Straub[26], Michael Wright[26], Simon Zwolinski[26], Zoe Allen[27], Birgitta Bernhard[27], Angela Brady[27], Claire Brooks[27], Louise Busby[27], Virginia Clowes[27], Neeti Ghali[27], Susan Holder[27], Rita Ibitoye[27], Emma Wakeling[27], Edward Blair[28], Jenny Carmichael[28], Deirdre Cilliers[28], Susan Clasper[28], Richard Gibbons[28], Usha Kini[28], Tracy Lester[28], Andrea Nemeth[28], Joanna Poulton[28], Sue Price[28], Debbie Shears[28], Helen Stewart[28], Andrew Wilkie[28], Shadi Albaba[29], Duncan Baker[29], Meena Balasubramanian[29], Diana Johnson[29], Michael Parker[29], Oliver Quarrell[29], Alison Stewart[29], Josh Willoughby[29], Charlene Crosby[30], Frances Elmslie[30], Tessa Homfray[30], Huilin Jin[30], Nayana Lahiri[30], Sahar Mansour[30], Karen Marks[30], Meriel McEntagart[30], Anand Saggar[30], Kate Tatton-Brown[30], Rachel Butler[31,32], Angus Clarke[31,32],

Sian Corrin[31,32], Andrew Fry[31,32], Arveen Kamath[31,32], Emma McCann[31,32], Hood Mugalaasi[31,32], Caroline Pottinger[31,32], Annie Procter[31,32], Julian Sampson[31,32], Francis Sansbury[31,32], Vinod Varghese[31,32], Diana Baralle[33,34,35], Alison Callaway[33,34,35], Emma J. Cassidy[33,34,35], Stacey Daniels[33,34,35], Andrew Douglas[33,34,35], Nicola Foulds[33,34,35], David Hunt[33,34,35], Mira Kharbanda[33,34,35], Katherine Lachlan[33,34,35], Catherine Mercer[33,34,35], Lucy Side[33,34,35], I. Karen Temple[33,34,35] & Diana Wellesley[33,34,35]

[11]North of Scotland Medical Genetics Service, NHS Grampian, Aberdeen, UK. [12]Northern Ireland Regional Genetics Centre, Belfast Health and Social Care Trust, Belfast City Hospital, Belfast, UK. [13]West Midlands Regional Genetics Service, Birmingham Women's NHS Foundation Trust, Birmingham Women's Hospital, Birmingham, UK. [14]Bristol Genetics Service, University Hospitals Bristol NHS Foundation Trust, St Michael's Hospital, Bristol, UK. [15]National Centre for Medical Genetics, Our Lady's Children's Hospital, Dublin, Ireland. [16]East of Scotland Regional Genetics Service, NHS Tayside, Ninewells Hospital, Dundee, UK. [17]Peninsula Clinical Genetics Service, Royal Devon and Exeter NHS Foundation Trust, Royal Devon & Exeter Hospital (Heavitree), Exeter, UK. [18]West of Scotland Regional Genetics Service, NHS Greater Glasgow and Clyde, Yorkhill Hospital, Glasgow, UK. [19]North East Thames Regional Genetics Service, Great Ormond Street Hospital for Children NHS Foundation Trust, Great Ormond Street Hospital, London, UK. [20]South East Thames Regional Genetics Centre, Guy's and St Thomas' NHS Foundation Trust, Guy's Hospital, London, UK. [21]Yorkshire Regional Genetics Service, Leeds Teaching Hospitals NHS Trust, Chapel Allerton Hospital, Leeds, UK. [22]Nottingham Regional Genetics Service, City Hospital Campus, Nottingham University Hospitals NHS Trust, Nottingham, UK. [23]Leicestershire Genetics Centre, University Hospitals of Leicester NHS Trust, Leicester Royal Infirmary, Leicester, UK. [24]Merseyside and Cheshire Genetics Service, Liverpool Women's NHS Foundation Trust, Royal Liverpool Children's Hospital Alder Hey, Liverpool, UK. [25]Manchester Centre for Genomic Medicine, Central Manchester University Hospitals NHS Foundation Trust, St Mary's Hospital, Manchester, UK. [26]Northern Genetics Service, Institute of Human Genetics, Newcastle upon Tyne Hospitals NHS Foundation Trust, Newcastle upon Tyne, UK. [27]North West Thames Regional Genetics Service, London North West University Healthcare NHS Trust, St Mark's Hospital, Harrow, UK. [28]Oxford Regional Genetics Service, Oxford Radcliffe Hospitals NHS Trust, Oxford, UK. [29]Sheffield Regional Genetics Services, Sheffield Children's NHS Trust, Sheffield, UK. [30]South West Thames Regional Genetics Centre, St George's Healthcare NHS Trust, St George's, University of London, London, UK. [31]Institute of Medical Genetics, University Hospital of Wales, Cardiff, UK. [32]Department of Clinical Genetics, Glan Clwyd Hospital, Rhyl, UK. [33]Wessex Clinical Genetics Service, University Hospital Southampton, Princess Anne Hospital, Southampton, UK. [34]Wessex Regional Genetics Laboratory, Salisbury NHS Foundation Trust, Salisbury District Hospital, Salisbury, UK. [35]Faculty of Medicine, University of Southampton, Southampton, UK.

