## [Peer Review File · Nature Communications]

Reviewer #1 (Remarks to the Author):

With interest I read the manuscript 'The contribution of X-linked coding variants to severe developmental disorders' by Martin and colleagues. The manuscript sets out to entangle one of clinical geneticists dogmas on the frequency of X-linked disorders in males and females based on genetic burden analyses. The manuscript is overall well-written, and provides proficiently sound conclusions based on the data. I have however a some questions and/or remarks.

- The introduction refers to Haldane's theory, and already provides some calculations, referring to the methods. Later in the results, the value 3.5 is used without further cross-referencing to the introduction where this number was explained. To me, the referral to the methods in the introduction made me wonder whether I had already missed something and in the results, I hence missed explanation. I would as such suggest to place the mathematic reasoning for the 18% to the results section of the manuscript, where it might better fits its purpose

- The final sentence of the second paragraph on page 4 ("It has been suggested that the 1.3-fold... Formally demonstrated"); I cannot follow the reasoning why this should be the last sentence of the paragraph? Perhaps provide this statement/hypothesis earlier in the manuscript?

- The authors use phenotypes of the male and female patients to assess any clinically statistical differences between the phenotypes observed. Can the authors comment on how many HPO terms per patient were on average (+range) were used in this analysis? Also, it seems that only 'a simple' scoring based on organ systems was used for this assessment. Whereas I understand this choice, I wonder whether the authors can comment on how they have treated the 'clinical severity' of different phenotypes within the same organ system. And similarly, if 'non-related' clinical phenotypes were scored in the same organ system, how they dealt with this information. Lastly, some clinical entities are related to one another, but not affecting the same organ system – how would this have influenced your data?

- In the results, the authors provide useful guidance on how to use CADD and MPC for missense variants on the X-chromosome. It would be good to re-iterate these findings and there implications in the discussion (paragraph on the diagnostic implications) for potential higher uptake in clinical diagnostic laboratories.

- In the UK biobank comparison to asses fitness consequences of heterozygous variants in X-linked recessive genes in females, the authors excluded all related, non-white British ancestry individuals. Whereas I understand that these type of stratifications might be needed if cohorts are sampled in two different ethnic backgrounds, I am not sure what the rational is in this case, as both the participants of the DDD study and those to the UK Biobank are samples from the same country – I would at least be surprised if there were no non-white British DDD participants given the historical migration background to the UK (Indian/Pakistani/Bengali).

- From the 13 carriers with a rare PTV in XLR genes in females in the UK BioBank, the authors conclude that females have nominally significant reduced number of children. They later couple this to the theory that this is because 'if there is an affected child in the family, they tend to be more hesitant for a next child'. Given that there is only nominal significance and absolute numbers are small, I wonder whether there could also be other explanations for these numbers. Do the authors know about (male) offspring of these 13 females? And of the males born to these women, how many were indeed affected, and how many of these women had another child after the affected son? (Or phrased differently: how often was the youngest son the affected child). Also, have the authors compared other traits between these women? Could it be that perhaps the 13 women with PTVs in the UK biobank are younger than those that do not have PTVs, and that these 13 women are therefore still likely to have (more) children? Or alternatively, that perhaps these 13 women have an higher educational attainment, and are therefore perhaps starting their families later than those without the PTVs?

- The small absolute numbers available for calculations as listed above are reflected by the large 95% confidence interval [19-83%]; with such large confidence intervals, almost 'any obtained calculation' 'overlaps with it'. This is also one of the limitations the authors note themselves in the discussion. Combined with the previous remarks on the UK biobank analysis, can the authors speculate on how this would have affected their data and conclusions. That is, I find it difficult to 'value' the conclusions based on these numbers.

(Note: from a biological perspective I believe the conclusions make sense, I just wonder whether these data proof the hypothesis)

- Can the authors provide more detailed information on the DDD cohort for 'point 5' under the selection criteria? I would potentially expect that if a large fraction would have one or several

affected family members, this would bias the cohort towards (X-linked?) recessive traits, and thus may as such impact the results obtained? My assumption is that the vast majority of the cohort fall in group iii of singletons, but it would be nice to specify this.

- The discussion reports on diagnostic implications, and focusses largely on the 'variants of unknown clinical significance (VUS)'. They report that information on pathogenicity can be further obtained by segregation analysis in (male member of the) families, which is hardly done outside of genetic services. To the best of my knowledge, most data in ClinVar etc are obtained from genetic services, so segregation would have been taken care of? Also, the implication that 'it is challenging to diagnose males with rare inherited missense variants in X-linked DD genes' is not a new observation.

- The authors calculated that ~1.4% of the DDD cohort has an X-linked diagnosis in an as-yet-undiscovered gene. Can the authors speculate on what this means in 'the number of X-linked genes the authors expect to be disease-causing' and also, to speculate on the distribution XLD/XLR?

Reviewer #3 (Remarks to the Author):

This manuscript focuses on analysis of X-linked variants in the DDD cohort and their contribution to disease in male vs. female probands.

X-linked genes represents a conundrum for genetic analysis, as they require a different mode of analysis from autosomal genes. Factors influencing interpretation of X-linked variants include the sex of the proband (haploid males vs. diploid females), random X-inactivation in female probands, the obligatory maternal X transmission for male probands - which is complicated by X-inactivation and/or carrier status of mothers and low spontaneous mutation rate in oocytes - and the relative dominant, semi-dominant or recessive status of alleles in either sex. Hence, X-linked variants are often neglected/ignored from large population datasets and few studies have analysed the pattern of X-linked variants in a systematic manner.

Moreover, another curiosity of X-biology is that it is well known that for many disorders there is a strong sex-bias with an excess of affected males (DD: 1.4-fold male excess; ASD: 4-fold male excess), which is often assumed to originate from the increased expressivity of the single X-allele in males.

However, in this study, by systematically analysing the burden of fully penetrant X-linked variants in a large cohort of 11,046 DDD patients, the authors show that, surprisingly, males and female probands carry a similar rate of X-linked causative variants (~6-7%). Hence, females are as often affected as males by X-linked deleterious variants - a finding that goes against a common assumption in clinical genetics (i.e. that females are largely spared by X-linked mutations, unless the mutation is lethal in boys) - and that in fact excess X-linked deleterious variants do not account for the observed male bias.

The authors show that most X-disease genes have probably already been discovered. They also consider whether phenotypic impact of variants and mode of inheritance (recessive, semi-dominant, dominant) of X-linked disease genes can be refined to further support interpretation of variants in male and female probands. Finally, to tackle the question of the male-bias in DD, they perform a PRS analysis of 216 males with 'suspected X-linked' inheritance (i.e. multiple affected males in the same family). This low-power analysis (variants with MAF>5% for relevant traits) did not provide an explanation for the increased male susceptibility to DD.

Overall, this focused study presents rigorous statistical analysis of a large dataset and has been performed in an original manner and to a high-standard. I believe that the results and the discussion presented here will be very valuable to many in the field of genetics and in clinical practice, where they will help refine the interpretation of X-linked variants from WES/WGS data. There are however some limitations to this study (DDD bias sampling, very large confidence intervals...) - which have been helpfully noted by the authors in the discussion - but overall the results are convincing and thought-provoking, as they challenge empirical assumptions in clinical

genetics.

I have very few comments (see below) and overall I have found the manuscript to be well-written, concise and informative.

1. One aspect that has not been addressed here, is the possible contribution of mosaicism to the X-linked burden. The authors found that ~41% (23-100%) of the X-linked burden in males is caused by de novo mutations. Given the low mutation rate in oocytes and the obligatory maternal origin of these mutations, was there any evidence in the sequence data of low maternal mosaicism that could explain this relatively 'high' contribution. Rahbari et al, (Nat Gen 2016) have previously shown that if a de novo mutation is maternal in origin, it is ~4x more likely to be mosaic than if it is paternal in origin.

Were there instances of post-zygotic mosaicism in male patients? Given the single X chromosome, this should be quite easy to call. For some genes, including MecP2 and some 'epilepsy' genes, it has been shown that germline and post-zygotic mosaicism are important contributor to disease.

2. From Fig 2, it is not clear whether, aside from MED12, there are other genes that may have been misclassified as XLR?

3. Page 10 – MED12 and classification of genes as XLR and XLD: Although it would be helpful to have a tighter classification of XLR and XLD genes to facilitate clinical interpretation of X-linked variants, the phenotype of mutations in X-genes in females will always be subject to non-penetrance - because of random X-inactivation, which effectively causes females to be mosaic for the mutation.

Even if we can gather information about the pattern on X-inactivation in an individual – and assuming it is accessible, in some of the relevant adult tissues (i.e. brain, as suggested here), it is clear that linking expressivity of a mutation to the pattern of X-inactivation will only ever be an inference – especially when dealing with "developmental disorders", which likely originate from cellular defects occurring early during embryogenesis when cells undergo complex morphogenetic rearrangements. Moreover, the majority of DDs are syndromic and do not only affect brain functions (ie Supp Table 1).

4. UK biobank data – 13 females (from a total of 18,632) with deleterious X-variants – this is a very small sample size (as acknowledged by the authors) but the analysis of reproductive fitness is nevertheless valuable. It may be helpful to know what are these 13 variants? Are these known disease variants? Are they in gnomAD?

5. Fig 1A shows that for females all variants were de novo (hatched). As I understand it, this is because only fully penetrant variants are considered. It may be worth stating this in the text.

6. Could the 23 significant genes (present on Figure 2) be highlighted in Sup Table 4

Response to reviewers

The reviewers' comments are in black below, with our responses interspersed in blue.

Note: In the course of tidying up the code to release with this paper, we made some minor improvements and fixed some small errors and inconsistencies (e.g. which version of the DDG2P list was used throughout). No findings changed in any meaningful way. The minor changes made to the manuscript have been detailed in an Appendix at the end of this response.

REVIEWER COMMENTS

Reviewer #1 (Remarks to the Author):

With interest I read the manuscript 'The contribution of X-linked coding variants to severe developmental disorders' by Martin and colleagues. The manuscript sets out to entangle one of clinical geneticists' dogmas on the frequency of X-linked disorders in males and females based on genetic burden analyses. The manuscript is overall well-written, and provides proficiently sound conclusions based on the data.

We thank the reviewer for these kind remarks.

I have however some questions and/or remarks.

- The introduction refers to Haldane's theory, and already provides some calculations, referring to the methods. Later in the results, the value 3.5 is used without further cross-referencing to the introduction where this number was explained. To me, the referral to the methods in the introduction made me wonder whether I had already missed something and in the results, I hence missed explanation. I would as such suggest to place the mathematic reasoning for the 18% to the results section of the manuscript, where it might better fits its purpose

This is a very reasonable point. We have rearranged the explanation to make this clearer, moving the mathematical reasoning from the Introduction to the Results.

- The final sentence of the second paragraph on page 4 ("It has been suggested that the 1.3-fold... Formally demonstrated"); I cannot follow the reasoning why this should be the last sentence of the paragraph? Perhaps provide this statement/hypothesis earlier in the manuscript?

In hindsight, we agree with the reviewer. We have moved this sentence to a more appropriate place in the final paragraph of the Introduction.

- The authors use phenotypes of the male and female patients to assess any clinically statistical differences between the phenotypes observed. Can the authors comment on how many HPO terms per patient were on average (+range) were used in this analysis?

We have added this information to the first paragraph of the Results: "Males had slightly more affected organ systems than females, although this was only nominally significant (mean and ranges: 3.55 [1-12] for males, 3.49 [1-11] for females; linear regression $p=0.049$; Supplementary Fig. 1)."

Also, it seems that only 'a simple' scoring based on organ systems was used for this assessment. Whereas I understand this choice, I wonder whether the authors can comment on how they have treated the 'clinical severity' of different phenotypes within the same organ system. And similarly,

if 'non-related' clinical phenotypes were scored in the same organ system, how they dealt with this information. Lastly, some clinical entities are related to one another, but not affecting the same organ system – how would this have influenced your data?

We have not attempted to rank phenotypes within the same organ system by phenotypic severity because this is somewhat subjective, and would not be feasible given the large number of probands and HPO terms we have (13,462 patients and 4,224 HPO terms observed in DDD). Similarly, for the same reason, we did not attempt to curate the phenotypes within organ systems as being 'non-related'. We also did not attempt to 'uniquify' related clinical entities that affect different organ systems. If we had done this, presumably it would have reduced the number of 'affected organ systems', although it seems unlikely this would have affected males versus females differently. Since the focus of our manuscript is primarily on the genetic burden analysis, we feel an in-depth re-curation of the HPO tree for these purposes is beyond its scope.

- In the results, the authors provide useful guidance on how to use CADD and MPC for missense variants on the X-chromosome. It would be good to reiterate these findings and their implications in the Discussion (paragraph on the diagnostic implications) for potential higher uptake in clinical diagnostic laboratories.

Thank you for this suggestion. We have added an extra sentence to that paragraph in the Discussion: "For example, considering all inherited missense variants absent from gnomAD in DDG2P genes, 13% [4%-22%] of these will be truly pathogenic, versus 53% [37-65%] of those with MPC score > 2, but applying the latter filter may lose about a third of diagnoses."

- In the UK biobank comparison to assess fitness consequences of heterozygous variants in X-linked recessive genes in females, the authors excluded all related, non-white British ancestry individuals. Whereas I understand that these type of stratifications might be needed if cohorts are sampled in two different ethnic backgrounds, I am not sure what the rationale is in this case, as both the participants of the DDD study and those to the UK Biobank are samples from the same country – I would at least be surprised if there were no non-white British DDD participants given the historical migration background to the UK (Indian/Pakistani/Bengali).

The rationale for restricting the UK Biobank samples to the white British was that we wanted to ensure that the variants retained in that analysis were truly rare and likely deleterious (since the variants would likely be enriched for false positives in this dataset). This is more difficult to do in ancestry groups not well represented in existing population studies of genetic variation. In response to the reviewer's comment, we repeated our UK Biobank analysis while including individuals of all ancestries. This added an additional four carriers of rare PTVs in consensus hemizygous DDG2P genes, bringing the total to seventeen. We estimated that they had 0.59 fewer children than non-carriers (95% CI 0.04 - 1.15 fewer, $p = 0.04$), which was very similar to the result with the white British individuals alone (difference of -0.55 children [95% CI -1.16 - 0.07], $p = 0.08$).

In DDD, about 85% of the samples are white British, but we retained all samples regardless of ancestry, since we were using the fathers as controls, and these are perfectly ancestry-matched to the offspring. While it is possible that some variants we included in the non-European subset of DDD might actually be more common in other populations, this would have been the case for the probands and fathers equally, so was less of a concern. We repeated all analyses in the European-only subset of DDD and none of our findings changed substantively, so we prefer to focus the manuscript on the full DDD dataset.

- From the 13 carriers with a rare PTV in XLR genes in females in the UK BioBank, the authors conclude that females have nominally significant reduced number of children. They later couple this

to the theory that this is because 'if there is an affected child in the family, they tend to be more hesitant for a next child'. Given that there is only nominal significance and absolute numbers are small, I wonder whether there could also be other explanations for these numbers. Do the authors know about (male) offspring of these 13 females? And of the males born to these women, how many were indeed affected, and how many of these women had another child after the affected son? (Or phrased differently: how often was the youngest son the affected child).

Unfortunately information about the number of male versus female children, and those children's phenotypes, is not available for the UK Biobank participants. We did look into the hospital episode statistics for the carrier individuals, but found that none of them had an ICD-10 code (principally in chapters XV and XVI) which could be plausibly associated with fetal abnormalities. One female did have a code for "Evidence of foetal stress", which does pass nominal statistical significance when considering the total number of unrelated white British ancestry females with this code in the UK Biobank ($n = 407$ individuals, Fisher's exact test $p = 0.03$). However, when considering all codes that are found among the carriers of PTVs in consensus hemizygous DD genes, no single ICD-10 code passed multiple testing correction for enrichment compared to the UK Biobank population as a whole (# codes = 110, significance threshold of $p < 4.5 \times 10^{-4}$).

Also, have the authors compared other traits between these women? Could it be that perhaps the 13 women with PTVs in the UK biobank are younger than those that do not have PTVs, and that these 13 women are therefore still likely to have (more) children? Or alternatively, that perhaps these 13 women have a higher educational attainment, and are therefore perhaps starting their families later than those without the PTVs?

We investigated if female hemizygous carriers were significantly younger than the UK Biobank population as a whole, but due to low numbers we believe this analysis is likely to be underpowered. The mean age of all females in the UK Biobank included in our analysis is 56.5. Of the 13 females who have PTVs in consensus hemizygous DD genes, 11 exceed this age (66, 63, 68, 66, 61, 66, 62, 61, 62, 61, and 59 years of age) and two are younger (50 and 42 years of age). While circumstantial, we consider it unlikely that this represents any age bias among carriers. We also note that, as indicated by the mean age of participants, the vast majority of women recruited to UK Biobank are either soon to be or past child-bearing age and thus age is unlikely to have a major impact on our findings.

To answer the reviewers' question as to educational attainment, we asked if PTV carriers completed college (i.e. higher education) at a rate different to that of non-carriers. 4/13 (30.8%) of PTV carrier females self-reported as completing college, which is almost identical to the overall proportion in the white British ancestry females from the entire UK Biobank cohort (30.5%). As such, while the overall numbers of PTV carriers is low, we likewise believe differential educational attainment is unlikely to play a role in our findings.

- The small absolute numbers available for calculations as listed above are reflected by the large 95% confidence interval [19-83%]; with such large confidence intervals, almost any obtained calculation overlaps with it. This is also one of the limitations the authors note themselves in the discussion. Combined with the previous remarks on the UK biobank analysis, can the authors speculate on how this would have affected their data and conclusions. That is, I find it difficult to 'value' the conclusions based on these numbers.

We agree with the reviewer that this particular result is very tentative. We have added a phrase into the Discussion to emphasise this: "We observed ~26% lower reproductive success (fertility ratio 0.742 [0.503-0.981]) in a small sample (N=13) of carrier females in UK Biobank compared to non-carriers, *but given the large confidence interval, this result should be treated with caution.*"

(Note: from a biological perspective I believe the conclusions make sense, I just wonder whether these data proof the hypothesis)

- Can the authors provide more detailed information on the DDD cohort for 'point 5' under the selection criteria? I would potentially expect that if a large fraction would have one or several affected family members, this would bias the cohort towards (X-linked?) recessive traits, and thus may as such impact the results obtained? My assumption is that the vast majority of the cohort fall in group iii of singletons, but it would be nice to specify this.

Unfortunately, the recruiting clinicians were not asked to specify directly which selection criteria were met by each patient. However, they did report whether the mother, father, siblings or other relatives were affected, and indeed the reviewer is correct: 73.5% of male probands and 79.0% of female probands had no affected family members. We have added a sentence about this to the first paragraph of the Results: "Males were more likely to have another affected family member than females (26.5% versus 21.0%; Fisher's exact test $p=5 \times 10^{-13}$)."

- The discussion reports on diagnostic implications, and focuses largely on the 'variants of unknown clinical significance (VUS)'. They report that information on pathogenicity can be further obtained by segregation analysis in (male member of the) families, which is hardly done outside of genetic services. To the best of my knowledge, most data in ClinVar etc are obtained from genetic services, so segregation would have been taken care of?

Data in ClinVar are largely derived from genetics laboratories and are often based on proband-only sequencing and analysis. As genomic diagnosis becomes mainstreamed to adult physicians and paediatricians, segregation studies are conducted less and less frequently as they are mainly undertaken by clinical genetic services.

Also, the implication that 'it is challenging to diagnose males with rare inherited missense variants in X-linked DD genes' is not a new observation.

We agree with the reviewer that this is not a new observation. However, we still feel that our study provides an important contribution by quantifying the positive predictive value of different types of variants in X-linked DD genes, which we hope will help guide clinical interpretation of variants.

- The authors calculated that ~1.4% of the DDD cohort has an X-linked diagnosis in an as-yet-undiscovered gene. Can the authors speculate on what this means in 'the number of X-linked genes the authors expect to be disease-causing' and also, to speculate on the distribution XLD/XLR?

We have done these kinds of analyses previously using simulations across all genes (<https://doi.org/10.1101/797787>) and for regulatory elements (<https://doi.org/10.1038/nature25983>). Our experience suggests that, given our sample size, applying similar methods to a single chromosome (chrX) would produce estimates with very wide confidence intervals. Furthermore, the simulations would require strong assumptions about penetrance and the relative contribution of *de novo* versus inherited variants in as-yet-undiscovered genes. Thus, we do not believe these investigations would be very informative.

Reviewer #3 (Remarks to the Author):

This manuscript focuses on analysis of X-linked variants in the DDD cohort and their contribution to disease in male vs. female probands.

X-linked genes represent a conundrum for genetic analysis, as they require a different mode of analysis from autosomal genes. Factors influencing interpretation of X-linked variants include the sex of the proband (haploid males vs. diploid females), random X-inactivation in female probands, the obligatory maternal X transmission for male probands - which is complicated by X-inactivation and/or carrier status of mothers and low spontaneous mutation rate in oocytes - and the relative dominant, semi-dominant or recessive status of alleles in either sex. Hence, X-linked variants are often neglected/ignored from large population datasets and few studies have analysed the pattern of X-linked variants in a systematic manner.

Moreover, another curiosity of X-biology is that it is well known that for many disorders there is a strong sex-bias with an excess of affected males (DD: 1.4-fold male excess; ASD: 4-fold male excess), which is often assumed to originate from the increased expressivity of the single X-allele in males.

However, in this study, by systematically analysing the burden of fully penetrant X-linked variants in a large cohort of 11,046 DDD patients, the authors show that, surprisingly, males and female probands carry a similar rate of X-linked causative variants (~6-7%). Hence, females are as often affected as males by X-linked deleterious variants – a finding that goes against a common assumption in clinical genetics (i.e. that females are largely spared by X-linked mutations, unless the mutation is lethal in boys) - and that in fact excess X-linked deleterious variants do not account for the observed male bias.

The authors show that most X-disease genes have probably already been discovered. They also consider whether phenotypic impact of variants and mode of inheritance (recessive, semi-dominant, dominant) of X-linked disease genes can be refined to further support interpretation of variants in male and female probands. Finally, to tackle the question of the male-bias in DD, they perform a PRS analysis of 216 males with ‘suspected X-linked’ inheritance (i.e. multiple affected males in the same family). This low-power analysis (variants with $MAF > 5\%$ for relevant traits) did not provide an explanation for the increased male susceptibility to DD.

Overall, this focused study presents rigorous statistical analysis of a large dataset and has been performed in an original manner and to a high-standard. I believe that the results and the discussion presented here will be very valuable to many in the field of genetics and in clinical practice, where they will help refine the interpretation of X-linked variants from WES/WGS data.

There are however some limitations to this study (DDD bias sampling, very large confidence intervals...) – which have been helpfully noted by the authors in the discussion – but overall the results are convincing and thought-provoking, as they challenge empirical assumptions in clinical genetics.

I have very few comments (see below) and overall I have found the manuscript to be well-written, concise and informative.

We thank Dr Goriely for this very positive response to our manuscript.

1. One aspect that has not been addressed here, is the possible contribution of mosaicism to the X-linked burden. The authors found that ~41% (23-100%) of the X-linked burden in males is caused by de novo mutations. Given the low mutation rate in oocytes and the obligatory maternal origin of these mutations, was there any evidence in the sequence data of low maternal mosaicism that could explain this relatively ‘high’ contribution. Rahbari et al, (Nat Gen 2016) have previously shown that if a de novo mutation is maternal in origin, it is ~4x more likely to be mosaic than if it is paternal in origin. Were there instances of post-zygotic mosaicism in male patients? Given the single X chromosome, this should be quite easy to call. For some genes, including MecP2 and some

‘epilepsy’ genes, it has been shown that germline and post-zygotic mosaicism are important contributors to disease.

Of the 127 *de novo* PTVs or missense/inframe mutations observed in males, eight (6%) appeared mosaic in mothers, and 12 (9%) post-zygotic mosaic in the probands. [We have added this to the second paragraph in the “X chromosome burden analysis” section of the Results.] The observation that 6% of *de novos* being parental mosaics is higher than the 0.5% we previously observed for autosomes (<https://pubmed.ncbi.nlm.nih.gov/31278258/>), as expected given the results in Rahbari *et al.* (Nat Gen 2016), since the X-linked *de novos* in males are, by definition, maternal in origin. The excess number (observed-expected) of LoF+missense/inframe *de novos* in males was 97.8, so the vast majority of this burden comes from constitutive *de novos*, with maternal mosaicism contributing <10%. In the female probands, we observed that three out of 381 *de novo* PTVs or missense/inframe variants were mosaic in the mothers. While the *de novo* mutations have not been phased so a large proportion will actually have arisen on the paternal haplotype, the comparison with the maternal *de novos* in males seems broadly consistent with the results from Rahbari *et al.*

2. From Fig 2, it is not clear whether, aside from MED12, there are other genes that may have been misclassified as XLR?

Indeed, *MED12* is the only gene for which we have strong evidence that it has been misclassified. We have now noted this in the Results: “The patterns of enrichment we observed were largely consistent with the inheritance modes previously reported for these genes, *with the exception of MED12 which we discuss below.*”

3. Page 10 – MED12 and classification of genes as XLR and XLD: Although it would be helpful to have a tighter classification of XLR and XLD genes to facilitate clinical interpretation of X-linked variants, the phenotype of mutations in X-genes in females will always be subject to non-penetrance - because of random X-inactivation, which effectively causes females to be mosaic for the mutation. Even if we can gather information about the pattern on X-inactivation in an individual – and assuming it is accessible, in some of the relevant adult tissues (i.e. brain, as suggested here), it is clear that linking expressivity of a mutation to the pattern of X-inactivation will only ever be an inference – especially when dealing with “developmental disorders”, which likely originate from cellular defects occurring early during embryogenesis when cells undergo complex morphogenetic rearrangements. Moreover, the majority of DDs are syndromic and do not only affect brain functions (ie Supp Table 1).

We agree with the reviewer on this and apologise that we were not explicit about this before. We have added an extra sentence into the Discussion to this effect: “Although our approach was designed to distinguish XLD from XLR genes, it is clear that more data will be needed from larger cohorts for a data-based classification of inheritance modes. *Furthermore, information in X-inactivation in females may help us to interpret inheritance modes, since this does contribute to penetrance in females, although accessibility of the relevant tissue is likely to be a challenge here.*”

4. UK biobank data – 13 females (from a total of 18,632) with deleterious X-variants – this is a very small sample size (as acknowledged by the authors) but the analysis of reproductive fitness is nevertheless valuable. It may be helpful to know what are these 13 variants? Are these known disease variants? Are they in gnomAD?

We agree with the reviewer and have now included a table (now Supplementary Table 4) which documents all thirteen variants identified as part of this analysis. As we have indicated in the legend, none of the variants are found in gnomAD v3 or in ClinVar.

5. Fig 1A shows that for females all variants were de novo (hatched). As I understand it, this is because only fully penetrant variants are considered. It may be worth stating this in the text.

This is correct. We have made this more explicit in the legend to Figure 1 and at the end of the first paragraph of the Results section on “X chromosome burden analysis”.

6. Could the 23 significant genes (present on Figure 2) be highlighted in Sup Table 4

We have done as the reviewer suggests.

Appendix: Additional minor changes made to the manuscript since the first submission

- The sample sizes should have been 7,844 males versus 5,618 females in DDD, 7,136 independent male probands, and 11,044 independent probands overall.
- We made a minor improvement in filtering which altered the number of *de novos* in a few genes.
- In Figure 1C, some estimates of attributable fraction were wrong due to a minor bug which has now been corrected.
- Some of the p-values and estimates of attributable fraction had been written incorrectly in Figure 2B and have been corrected.
- The TADA parameters used were slightly different from those previously given in the Methods and Supplementary Table 7.
- In the section on estimating the fertility ratio in UK Biobank, the upper bound on the 95% confidence interval for the expected fraction of de novos has been changed from 40.2% to 39.8%.
- The fraction of the burden driven by PTVs has been changed from 38.0% to 38.9%.
- The attributable fraction for X-linked coding variants in males with suspected X-linked inheritance has been changed from 16.2% to 16.4%.
- The ClinVar analysis was redone using an updated version of DDG2P which slightly changed the numbers there.

Reviewer #1 (Remarks to the Author):

I would to thank the authors for their extensive answers and analyses performed to address the questions raised. I would also like to compliment them on their achievements, and enjoyed reading the manuscript as is. I have no further queries for the authors to comment on.

Lisenka Vissers

Reviewer #3 (Remarks to the Author):

The authors have toughfully address all my comments. I believe this is a thorough study that will benefit the field.